# Re-ranking Reasoning Context with Tree Search Makes Large Vision-Language Models Stronger

**Qi Yang** [1 2 3]   **Chenghao Zhang** [3]   **Lubin Fan** [3]   **Kun Ding** [2]   **Jieping Ye** [3]   **Shiming Xiang** [1 2]

https://github.com/yannqi/RCTS-RAG

## Abstract

Recent advancements in Large Vision Language Models (LVLMs) have significantly improved performance in Visual Question Answering (VQA) tasks through multimodal Retrieval-Augmented Generation (RAG). However, existing methods still face challenges, such as the scarcity of knowledge containing reasoning examples and erratic responses from retrieved knowledge. To address these issues, in this study, we propose a multimodal RAG framework, termed RCTS, which enhances LVLMs by constructing a **R**easoning **C**ontext-enriched knowledge base and a **T**ree **S**earch re-ranking method. Specifically, we introduce a self-consistent evaluation mechanism to enrich the knowledge base with intrinsic reasoning patterns. We further propose a Monte Carlo Tree Search with Heuristic Rewards (MCTS-HR) to prioritize the most relevant examples. This ensures that LVLMs can leverage high-quality contextual reasoning for better and more consistent responses. Extensive experiments demonstrate that our framework achieves state-of-the-art performance across multiple VQA datasets, significantly outperforming both In-Context Learning (ICL) and Vanilla-RAG methods. It highlights the effectiveness of our knowledge base and re-ranking method in improving LVLMs.

## 1. Introduction

*"One example speaks louder than a thousand words."*

This work was done when Qi Yang was an intern at Alibaba Cloud Computing. [1]School of Artificial Intelligence, University of Chinese Academy of Sciences, China [2]MAIS, Institute of Automation, Chinese Academy of Sciences, China [3]Alibaba Cloud Computing, China. Correspondence to: Lubin Fan <lubin.flb@alibaba-inc.com>, Kun Ding <kun.ding@ia.ac.cn>.

*Proceedings of the $42^{nd}$ International Conference on Machine Learning*, Vancouver, Canada. PMLR 267, 2025. Copyright 2025 by the author(s).

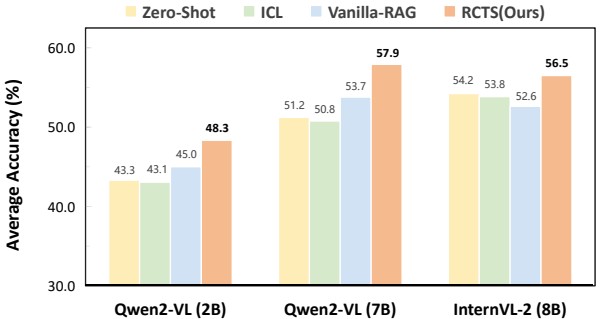

*Figure 1.* Comparison of various methods built on different LVLMs across multiple reasoning datasets. Our proposed RCTS framework demonstrates substantial performance gains over conventional Zero-Shot and Vanilla-RAG (Lin et al., 2024b) methods.

Recently, large vision language models (LVLMs) (Achiam et al., 2023; Bai et al., 2023; Chen et al., 2024) exhibit remarkable efficacy across diverse visual question answering (VQA) tasks, being capable of processing multiple images concurrently and, furthermore demonstrating the ability for in-context learning (ICL) (Alayrac et al., 2022; Wang et al., 2024b). These capabilities facilitate the application of multimodal retrieval-augmented generation (RAG) (Gao et al., 2023), a training-free approach that involves augmenting the input prompt by retrieving relevant multimodal corpus from an external knowledge base through semantic similarity calculation. This approach demonstrates that introducing external knowledge effectively reduces the probability that LVLMs generate incorrect content.

Existing LVLMs (Achiam et al., 2023; Bai et al., 2023; Chen et al., 2024) are prone to hallucination issues (Huang et al., 2023), which manifest in two primary forms: generating factual inconsistent with real-world facts (*e.g.*, misstating historical events or political news), and producing erratic responses misaligned with user instructions or questions (*e.g.*, failing to answer user queries). To mitigate factual inconsistencies, existing multimodal RAG methods (Chen et al., 2022; Caffagni et al., 2024; Yan & Xie, 2024) leverage external knowledge (*e.g.*, Wikipedia or Web Search) to transform LVLMs' responses from *unknown* (lacking factual ground-

ing) to *known* (factually supported). However, addressing instruction misalignment poses a distinct challenge. An intuitive approach is to enhance user prompts by prepending few-shot example pairs through in-context learning (Alayrac et al., 2022). While effective, manual curation of such examples limits scalability.

For issue instruction misalignment, a compelling question arises as to whether multimodal RAG can be integrated into in-context learning, transitioning LVLMs' responses from merely *known* to better *understood* (know-how reasoning) by prepending it with retrieved examples. Specifically, it can achieve more reliable responses by retrieving and reasoning over similar examples through in-context learning. However, several possible challenges hinder the practical application of multimodal RAG for addressing this question: i) The retrieved sample question-answer pairs are formatted in a rigid, formulaic manner (*e.g.*, 'The answer is A' for multiple-choice questions), which limits the LVLMs to capture underlying logical patterns. This inspires us to build a more comprehensive knowledge base with reasoning contexts. ii) Retrieved examples may not consistently result in positive outcomes, due to the inherent limitations of in-context learning and the diversity of users' queries, which deserves more discussion. Hence, the focus here centers on instruction misalignment with two main aspects: Firstly, the construction of the knowledge base with reasoning contexts to optimally enhance generation and facilitate in-context learning. Secondly, the strategic re-ranking of retrieved examples to prioritize more suitable samples, thereby promoting efficient and accurate response generation.

In this study, we propose a multimodal RAG framework with **R**easoning **C**ontext and **T**ree **S**earch, named **RCTS**, aiming at constructing a comprehensive knowledge base with reasoning contexts and optimizing the order of contextual examples to improve the question answering performance of LVLMs. For our knowledge base component, we introduce an automated reasoning contexts generation method for question-answer pairs, which helps LVLMs acquire intrinsic reasoning patterns. For the proposed multimodal RAG framework, our method begins with hybrid retrieval for an initial sampling from the knowledge base. Subsequently, we employ a re-ranking mechanism to organize the retrieved samples, enhancing the efficacy of in-context learning. The re-ranked Top-$K$ samples with the generated reasoning contexts are then concatenated with the user's question to facilitate optimal answer generation by LVLMs. Regarding the re-ranking process, we propose a tree search approach with heuristic rewards to re-order the retrieved samples. This ensures the identification and prioritization of the most beneficial contextual examples for the final generation phase, thereby enhancing the overall answer quality. Besides, the reasoning contexts we generated before also allows for a quantitative assessment of the potential benefits offered by the retrieved samples, reinforcing the efficacy of our tree search method.

To validate the effectiveness of our proposed method, we conduct extensive experiments across multiple reasoning VQA datasets, including ScienceQA (Lu et al., 2022), MMMU (Yue et al., 2024), and MathV (Wang et al., 2024a). Our method also excels in non-reasoning VQA datasets such as VizWiz (Gurari et al., 2018) and VSR-MC (Liu et al., 2023). As depicted in Fig. 1, across various sizes and types of LVLMs, our proposed approach significantly outperforms the zero-shot baseline. Besides, compared to the strategy of randomly selecting examples as context, *i.e.*, ICL, our method yields an average of 3% improvement, demonstrating that our framework elevates LVLMs from mere *known* to better *understood*. Additionally, compared to Vanilla-RAG, our method surpasses performance by more than 3% on all models (4.2% on Qwen2-VL (7B), 3.9% on InternVL-2 (8B)), indicating that the knowledge base with reasoning contexts and the tree search with answer heuristic rewards effectively re-rank examples that enhance answer accuracy. Qualitative analysis further corroborates the efficacy of our method.

Our contributions are summarized as follows:

- We introduce a multimodal RAG framework, termed RCTS, to enhance LVLMs by constructing a comprehensive knowledge base with reasoning contexts and re-ranking for highly relevant contexts.

- We develop an automatically constructed reasoning context mechanism grounded in VQA pairs to construct the knowledge base with reasoning contexts, and further propose a tree search strategy with answer heuristic rewards for re-ranking retrieved samples.

- Experiments show that our method achieves significant performance improvements on multiple VQA datasets, demonstrating the effectiveness of the reasoning context and the proposed re-ranking mechanism.

## 2. Related Work

**Large Visual Language Models.** Large Visual Language Models (LVLMs) have emerged as a significant research focus, leveraging the capabilities of powerful Large Language Models (LLMs) (Touvron et al., 2023; Jiang et al., 2023; Yang et al., 2024; Abdin et al., 2024) to tackle vision-language tasks. These versatile LVLMs demonstrate exceptional performance, particularly in visual question-answering (VQA) tasks (Team et al., 2023; Achiam et al., 2023; Bai et al., 2023; Liu et al., 2024), pointing toward a promising avenue for achieving artificial general intelligence. Nevertheless, these models face challenges with knowledge-based VQA due to issues such as hallucina-

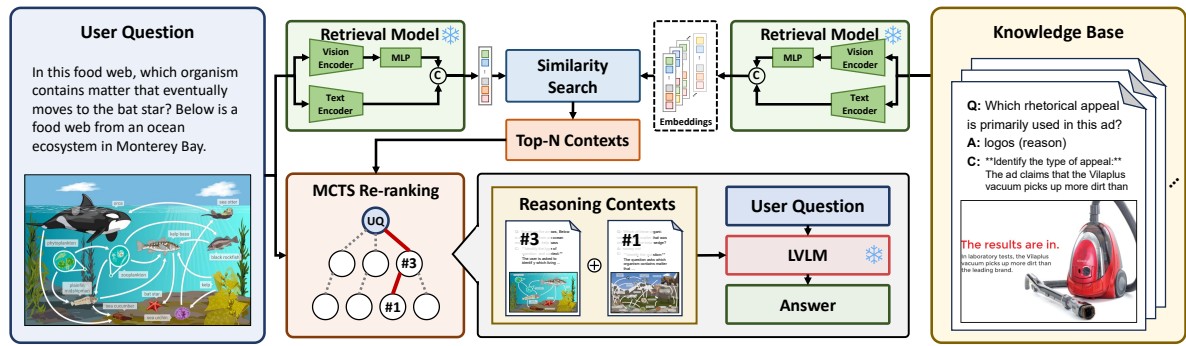

*Figure 2.* Overview of the proposed framework. RCTS adopts a novel multimodal retrieval-augmented generation framework specifically for visual question answering tasks. Aiming at enhancing the capabilities of the large vision-language models, our method consists of three components. (1) We construct a knowledge base with reasoning contexts by a self-consistent evaluation mechanism. (2) To support the multimodal knowledge base, we employ a hybrid embedding strategy for relevant samples retrieval. (3) Given the uncertainty of the retrieved samples, we propose an improved Monte Carlo Tree Search algorithm with heuristic rewards for sample re-ranking.

tions—where responses are generated from nonexistent content—and inherent biases (Li et al., 2023). Additionally, the lack of efficient knowledge retrieval mechanisms impedes their ability to integrate external knowledge bases for reasoning (Caffagni et al., 2024). In this study, we investigate strategies for constructing comprehensive external knowledge bases to augment the capabilities of LVLMs.

**Multimodal In-context Learning.** Multimodal in-context learning exemplifies a paradigm in which model weights remain unchanged, and improves output quality by adjusting the model's input (Dong et al., 2022; Alayrac et al., 2022; Han et al., 2023). A typical in-context learning prompt comprises two elements: demonstrations and new queries. Demonstrations involve multiple VQA pairs, each comprising a complete question accompanied by visual information and its corresponding answer. In contrast, new queries consist of questions posed to the model. Leveraging the emergent capabilities of LVLMs, these models can reference demonstrations to some extent to address new questions (Zhao et al., 2023; Zhang et al., 2024b). With the benefit of not requiring fine-tuning model parameters, in-context learning has emerged as a favored paradigm for applying LVLMs. In this study, we construct the reasoning context as an integral part of the context based on VQA pairs, to enrich the reasoning knowledge of the context.

**Multimodal Retrieval-augmented Generation.** While RAG is well-established in LLMs, its application within LVLMs remains relatively underexplored. Systems such as KAT (Gui et al., 2021), REVIVE (Lin et al., 2022), and RE-VEAL (Hu et al., 2023) show promise in addressing queries involving common-sense reasoning, yet they struggle with more complex, knowledge-intensive tasks like Encyclopedic VQA (E-VQA) (Mensink et al., 2023) and Infoseek (Chen et al., 2023). These limitations are largely due to their constrained ability to fetch and integrate precise information

from expansive encyclopedic knowledge bases. RATP (Pouplin et al., 2024) leverages MCTS and RAG to enhance the self-reflection and self-critique capabilities across numerous private healthcare documents. EchoSight (Yan & Xie, 2024) attempts to address these challenges through a two-stage process, combining visual-only retrieval and multimodal reranking, thereby enhancing the alignment between retrieved textual knowledge and visual content. However, this method risks losing the association and intrinsic knowledge of visual text due to the conversion of visual information into text. In contrast, our approach considers multimodal information in both the retrieval and reranking stages, thereby preserving the integrity of the knowledge base information more effectively.

## 3. Methodology

Humans always learn by examples. This cognitive process can be conceptualized as exploring isomorphic structures across diverse examples, thereby improving the extraction of heuristic insights (Van Gog & Rummel, 2010). Drawing inspiration from this cognitive paradigm, we hypothesize that LVLMs can similarly benefit from contextually relevant examples for in-context learning.

### 3.1. Problem Statement

Existing multimodal retrieval-augmented generation (RAG) techniques (Yan & Xie, 2024; Li et al., 2024) primarily address open-domain questions that LVLMs fail to answer without an external knowledge base. In contrast, we focus on scenarios where user queries fall within the scope of LVLMs' capabilities, albeit with potential inaccuracies. As shown in Fig. 2, LVLMs can take advantage of relevant examples retrieved from the knowledge base to obtain more precise and reliable responses.

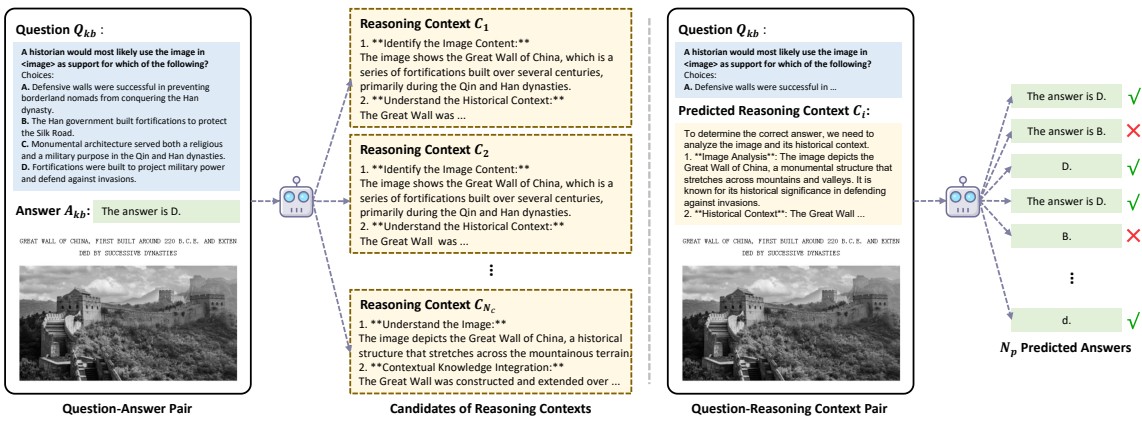

*Figure 3.* Illustration of Reasoning Context Generation. The generation method consists of two steps: (a) Utilizing the question-answer pairs from the knowledge base to generate self-consistent reasoning context. (b) Validating the predicted answer based on the quantitative evaluation for optimal reasoning context selection.

**Knowledge Base.** We define the knowledge base consisting of $M$ visual question-answer pairs, denoted as $\mathcal{D}_{KB} = \{x_i\}_{i=1}^{M}$. Each $x_i$ encompasses an image $I_i$, a question $Q_i$, its corresponding reference answer $A_i$, and an associated reasoning context $C_i$ (See Sec. 3.2). Formally, this can be expressed as $x_i := (I_i, Q_i, A_i, C_i)$.

**Goal.** Our framework leverages the user's query $(I_u, Q_u)$ to retrieve $K$ pertinent question-answer pairs $X_{ret} = (x_1, x_2, ..., x_K)$ from the existing knowledge base $\mathcal{D}_{KB}$. Subsequently, the framework generates predicted answer $\tilde{y}$ using large vision-language model $\mathcal{G}$:

$$\tilde{y} \sim \mathcal{G}\left([I_u; Q_u; X_{ret}]\right), \ X_{ret} \subseteq \mathcal{D}_{KB}. \quad (1)$$

The goal is to develop a multimodal RAG framework that effectively integrates retrieved information with in-context learning to make the predicted answer $\tilde{y}$ align closely with the ground-truth response. It is worth noting that our framework is training-free and can be adaptively extended to multiple domains by simply expanding the knowledge base.

### 3.2. Reasoning Context with Self-Consistent Evaluation

Existing knowledge bases usually include visual question-answer pairs without detailed reasoning procedures, which makes it difficult to provide valuable context for responses even if relevant examples are retrieved. To alleviate this issue, drawing from Auto-CoT (Zhang et al., 2022), we propose a method capable of automatically generating reasoning contexts for visual question-answer pairs to enhance contextual information during generation. We leverage a self-consistency mechanism of LVLMs to generate candidate reasoning contexts and utilize mutual answer prediction for reasoning context verification.

Specifically, as in Fig. 3 (a), given a question-answer pair $(Q_{kb}, A_{kb})$ from the knowledge base, $N_c$ candidate reasoning contexts are generated first by LVLMs, denoted as $\{C_i\}_{i=1}^{N_c}$. In Fig. 3 (b), $N_p$ predicted answers are generated by combining the question $Q_{kb}$ with each $\{C_i\}_{i=1}^{N_c}$. These predicted answers are evaluated with the ground truth answer $A_{kb}$ to obtain a set of prediction scores $\{\text{Score}_i\}_{i=1}^{N_c}$. Finally, the candidate reasoning context with the highest score is selected as the associated reasoning context.

### 3.3. Knowledge Retrieval with Hybrid Embeddings

As shown in Fig. 2, considering that both the knowledge base and user queries contain multimodal information, we employ hybrid-modal retrieval approaches rather than relying solely on a single modality. Following (Lin et al., 2023; 2024b), given user's query consisting of an image $I_u$ and a question $Q_u$, we first use a text encoder $\mathcal{F}_L$ and an image encoder $\mathcal{F}_I$ with linear function to obtain their embeddings with the same dimension $d$. The formulation is as follows:

$$\begin{aligned} \mathbf{E}_{T_u} &= \mathcal{F}_L(Q_u) \in \mathbb{R}^{l_{T_u} \times d}; \\ \mathbf{E}_{I_u} &= \mathcal{F}_I(I_u) \in \mathbb{R}^{l_{I_u} \times d}, \end{aligned} \quad (2)$$

where $l_{T_u}$ and $l_{I_u}$ denote the total number of tokens of question $Q_u$ and image $I_u$, respectively.

To enable hybrid-modal retrieval, all token-level embeddings are concatenated for retrieval, *i.e.*, $\mathbf{E_u} = [\mathbf{E}_{T_u}, \mathbf{E}_{I_u}] \in \mathbb{R}^{(l_{T_u}+l_{I_u}) \times d}$. Similarly, to maintain consistency with user queries, we utilize the same text questions and images, excluding answers from the knowledge base for the retrieval process. The knowledge base hybrid embeddings are defined as:

$$\mathbf{E_{KB}} = \{\mathbf{E_i}\}_{i=1}^{M} = \{[\mathbf{E}_{T_i}, \mathbf{E}_{I_i}]\}_{i=1}^{M}. \quad (3)$$

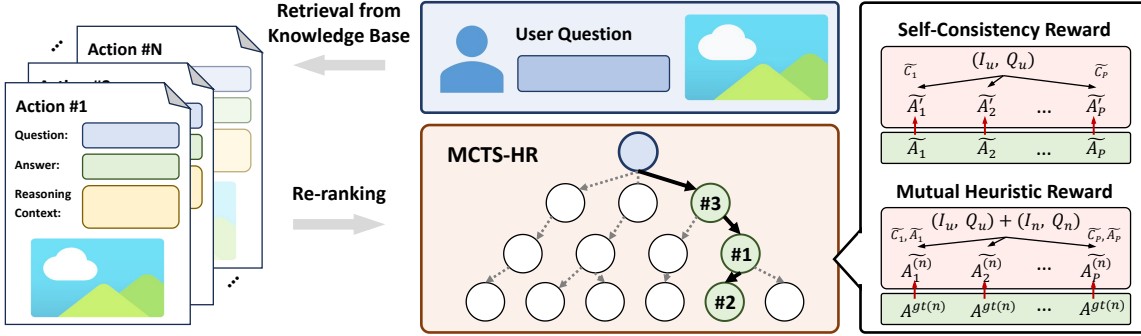

*Figure 4.* Illustration of our Monte Carlo Tree Search with Heuristic Rewards (MCTS-HR). To address the user's query, we initially retrieve Top-$N$ samples as candidate actions, which are subsequently selected through our MCTS-HR for sample re-ranking. Additionally, we propose a heuristic reward strategy that incorporates two key components, a self-consistency heuristic reward, and a mutual heuristic reward, to optimize the reward function within the MCTS framework.

Finally, we compute the relevance score $r$ between user queries embeddings $\mathbf{E_u}$ and each knowledge base embeddings $\mathbf{E_{KB}}_i$ as follows:

$$r(\mathbf{E_u}, \mathbf{E_i}) = \sum_{j=1}^{l_u} \max_{k=1}^{l_i} \mathbf{E_u}_j \mathbf{E_i}_k^\top, \qquad (4)$$

where $l_u = l_{T_u} + l_{I_u}$ and $l_i$ represent the number of tokens in hybrid embeddings, respectively. Therefore, the final relevance scores $\mathbf{r}(\mathbf{E_u}, \mathbf{E_{KB}}) = \{[r(\mathbf{E_u}, \mathbf{E_i})]\}_{i=1}^M$. The Top-$N$ pertinent question-answer pairs $X_{ret-N} = (x_1, x_2, ..., x_N)$ are chosen by relevance scores $\mathbf{r}$.

### 3.4. Re-ranking by Tree Search with Heuristic Rewards

This stage aims to re-rank the retrieved samples for selecting the most pertinent samples as the context prompt, facilitating efficient and accurate answer generation. Specifically, we adopt the Monte Carlo Tree Search (MCTS) methodology (Browne et al., 2012), a technique primarily employed in designing game-playing bots, to enhance the sample selection and re-ranking processes. Since MCTS can effectively balance exploring diverse samples and exploiting high-quality ones through simulated trajectories, solving combinatorial optimization in context selection, we formulate the task as a sequential decision-making problem and propose a Monte Carlo Tree Search with Heuristic Reward (MCTS-HR) strategy, as shown in Fig. 4. A detailed workflow of our proposed MCTS-HR is provided in Appendix A.

Formally, we initialize a root node with a zero-shot response derived from the user's query. Then, existing nodes are ranked and selected for expansion using a greedy sampling strategy based on visit times $N(a)$ and node values $Q(a)$. During node expansion, action is sampled from an action space constructed from the retrieval samples $X_{ret}$. When the maximum depth is reached, the algorithm performs a simulation by concatenating actions and the user query to

form a $K$-shot prompt, generating a response for evaluation. This response is then assessed with a reward function $\mathcal{R}$, and the reward value $Q$ is backpropagated to update the tree's value information. Following the standard MCTS procedure, the upper confidence bound for trees (UCT) values of all nodes are then updated to guide further exploration. The algorithm iterates through these stages, re-ranking retrieved samples and refining responses until a termination condition, such as a maximum number of rollouts (referring to the number of simulations) or an early stopping strategy, is met. Below, we introduce the key elements of our algorithm.

**Actions Construction and Selection.** Unlike most MCTS-based methods (Zhang et al., 2024a; Qi et al., 2024) in LLMs that rely on human-defined prompts as actions to construct the tree, our approach employs question-answer pairs retrieved from the external knowledge base as candidate actions $\mathcal{A}$. Formally, we employ hybrid embeddings to retrieve the $N$ most relevant question-answer pairs, where $N \gg K$, and constitute the full action space:

$$\mathcal{A} = \{[x_1, s_1], [x_2, s_2], ..., [x_N, s_N]\}, \qquad (5)$$

where $x_i = (I_i, Q_i, A_i, C_i)$, $s_i$ denotes the normalized similarity score between the retrieved pair $x_i$ and user's query.

During the node expansion stage, let $\mathcal{C} \subset \mathcal{A}$ be the set of actions that have already been selected (*i.e.*, actions in the parent node). The remaining valid actions available for sampling are $\mathcal{A}_{valid} = \mathcal{A} \setminus \mathcal{C}$. Then, MCTS takes an action $a_i \sim P(a_i)$ from the action space $\mathcal{A}_{valid}$ using similarity-based probability distribution:

$$P(a_i) = \frac{s_i}{\sum_{j, a_j \in \mathcal{A}_{valid}} s_j}, \qquad (6)$$

where the selected action $a_i$ serves as the re-ranked example $x_i$, and this process continues iteratively until it reaches its maximum depth $K$, thereby completing a branch of the

MCTS. Finally, the sequence of $K$ actions extracted from the current branch is concatenated with the user's query to form a $K$-shot prompt, thereby completing a branch simulation and obtaining the response for this branch.

**Self-Consistency and Mutual Heuristic Rewards.** Another critical component of MCTS is the reward function $\mathcal{R}$, which evaluates the value of each action and directs the tree expansion. Unlike the traditional MCTS-based LLM methods (Qi et al., 2024; Zhang et al., 2024a), which directly uses a language model as reward function $\mathcal{R}$ to score the node response, we propose a *self-consistency heuristic reward strategy* to get the self-reward value $Q_S$ alongside a *mutual heuristic reward strategy* to get the mutual-reward value $Q_M$ based on the in-context consistency.

For *self-consistency heuristic reward strategy*, assuming that the predicted $K$-shot response $\tilde{y}_i$ at branch $i$ is denoted as $\tilde{y}_i = (\tilde{A}_i, \tilde{C}_i)$. The user questions $(I_u, Q_u)$ and the prediction $\tilde{C}_i$ are concatenated to generate multiple answers $\{A_i^{(n)}\}_{n=1}^{N_s}$. In theory, these answers $A_i^{(n)}$ should be consistent with the originally predicted answers $\tilde{A}_i$. According to the above heuristic rules, the self-reward value $Q_{S,i}$ can be expressed as follows:

$$Q_{S,i} = \frac{1}{N_s} \sum_{n=1}^{N_s} \mathcal{R}\left(\tilde{A}_i'^{(n)}, \tilde{A}_i\right), \qquad (7)$$

where $A_i^{(n)} \sim \mathcal{G}\left([I_u; Q_u; \tilde{C}_i], n\right)$, $\mathcal{G}$ represents the large vision language model, $\mathcal{G}(\cdot, n)$ denotes the random seed in answer generation. Reward function $\mathcal{R}$ is calculated through the rule-based evaluator.

For *mutual heuristic reward strategy*, we posit that if the answer to a question is correct, it will positively contribute to other questions, and vice versa. Therefore, we greedily pick $N_m$ samples $\{(I_n, Q_n)\}_{n=1}^{N_m}$ from the actions space $\mathcal{A}$ to serve as subsequent mutual heuristic samples. For branch $i$, we utilize the user's question and the predicted response $\tilde{y}_i$ as contextual prompts, with the selected $N_m$ samples' questions as the reference question and its corresponding answer as the ground truth answer $A_i^{\text{gt}(n)}$. The predicted answer $\tilde{A}_i^{(n)}$ for the reference question should be consistent with the ground truth answer. Thus, the mutual-reward value $Q_{M,i}$ can be represented as:

$$Q_{M,i} = \frac{1}{N_m} \sum_{n=1}^{N_m} \mathcal{R}\left(\tilde{A}_i^{(n)}, A_i^{\text{gt}(n)}\right), \qquad (8)$$

where $\tilde{A}_i^{(n)} \sim \mathcal{G}\left([I_u; Q_u; \tilde{y}_i; I_n; Q_n]\right)$. And the final reward value $Q_i$ for each branch $i$ consists of self-reward value $Q_{S,i}$ and mutual-reward value $Q_{M,i}$ with a weighted summation as:

$$Q_i = \alpha \cdot Q_{S,i} + (1 - \alpha) \cdot Q_{M,i}, \qquad (9)$$

*Table 1.* Statistics of multiple VQA datasets divided into evaluation set and knowledge base.

| Evaluation Set | | Knowledge Base | |
|---|---|---|---|
| Name | Size | Name | Size |
| ScienceQA$_{\text{test}}$ | 4241 | ScienceQA$_{\text{trainval}}$ | 16967 |
| MMMU-Dev | 150 | MMMU-Val | 900 |
| MathV$_{\text{testmini}}$ | 304 | MathV$_{\text{test}}$ | 2736 |
| VizWiz$_{\text{val}}$ | 4319 | VizWiz$_{\text{train}}$ | 20523 |
| VSR-MC$_{\text{test}}$ | 1181 | VSR-MC$_{\text{trainval}}$ | 4440 |

* No identical samples in evaluation set and knowledge base.

where $\alpha$ is a weighting parameter that controls the importance of the self-reward and mutual-reward values. More details in Section 4.4.

**Reward Backpropagation.** After obtaining the reward value $Q$, we then propagate this reward value to its parent and ancestor nodes. Formally, if the reward value of any element in the child node set Children($p$) changes, the reward value of the parent node $Q(p)$ is updated to:

$$Q'(p) = \frac{1}{2}\left(\frac{Q(p) \cdot N(p) + Q(c)}{N(p) + 1} + \max_{i \in \text{Children}(p)} Q(i)\right), \qquad (10)$$

where $N(p)$ denotes visit times of the parent node $p$. $Q(c)$ represents the reward value of the changed child node $c$. $\max_{i \in \text{Children}(p)} Q(i)$ represents the highest quality value among all child nodes of parent node $p$.

This formula takes into account not only the reliability of the answers of all child nodes in the parent node $p$, but also the reward value of the answer of the most outstanding child.

## 4. Experiments

### 4.1. Datasets

In our experimental benchmark, we carry out comprehensive experiments with three common reasoning VQA datasets in extensive domains, including **ScienceQA** (Lu et al., 2022), **MMMU** (Yue et al., 2024) and **MathV** (Wang et al., 2024a). Additionally, we compare methods on simpler, non-reasoning VQA datasets using **VizWiz** (Gurari et al., 2018) and **VSR-MC** (Liu et al., 2023). Following the original splits of these VQA datasets, we construct the knowledge base with the training set and build the evaluation set with the testing set, respectively. Tab. 1 presents the size statistics of the knowledge base and the evaluation set. Please refer to Appendix B for details and examples of the datasets.

### 4.2. Implementation Details

The proposed framework is applicable to mainstream LVLMs, thus we evaluate our method on various LVLMs across different scales and types, such as

*Table 2.* Comparison results using various LVLMs across different sizes and types on the ScienceQA, MMMU, and MathV datasets.

| Datasets | Knowledge Base | Methods | Large Vision Language Models | | |
|---|---|---|---|---|---|
| | | | Qwen2-VL (2B) | Qwen2-VL (7B) | InternVL-2 (8B) |
| ScienceQA$_{test}$ | ScienceQA$_{trainval}$ | Zero-Shot | 67.18 | 80.33 | 93.00 |
| | | ICL (random retrieval) | 70.10 | 81.63 | 93.14 |
| | | Vanilla-RAG (top retrieval) | 71.94 | 86.68 | 92.78 |
| | | **RCTS (ours)** | **78.99** | **91.44** | **94.20** |
| MMMU-Dev | MMMU-Val | Zero-Shot | **44.00** | 51.33 | 48.00 |
| | | ICL (random retrieval) | 41.33 | 47.33 | 47.33 |
| | | Vanilla-RAG (top retrieval) | 42.67 | 50.00 | 46.67 |
| | | **RCTS (ours)** | **44.00** | **53.33** | **51.33** |
| MathV$_{testmini}$ | MathV$_{test}$ | Zero-Shot | 18.75 | 22.04 | 21.71 |
| | | ICL (random retrieval) | 17.76 | 23.35 | 21.05 |
| | | Vanilla-RAG (top retrieval) | 20.39 | 24.67 | 18.42 |
| | | **RCTS (ours)** | **22.04** | **28.95** | **24.01** |

*Table 3.* Comparison results on VizWiz and VSR-MC datasets.

| Datasets | Knowledge Base | Methods | Qwen2-VL (7B) |
|---|---|---|---|
| VizWiz$_{val}$ | VizWiz$_{train}$ | Zero-Shot | 66.49 |
| | | ICL | 69.01 |
| | | Vanilla-RAG | 69.89 |
| | | **RCTS (ours)** | **71.50** |
| VSR-MC$_{test}$ | VSR-MC$_{trainval}$ | Zero-Shot | 51.22 |
| | | ICL | 52.32 |
| | | Vanilla-RAG | 52.92 |
| | | **RCTS (ours)** | **55.97** |

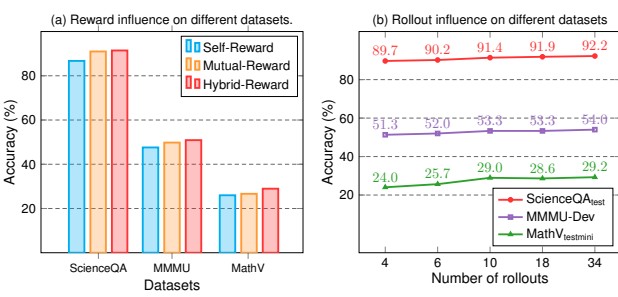

*Figure 5.* (a) Ablation of reward strategy on different datasets. (b) Ablation of rollouts on different datasets.

*Table 4.* Ablation study of key components in our method, where Rea. Con. represents Reasoning Context.

| Module | | ScienceQA | MMMU | Math-V |
|---|---|---|---|---|
| MCTS | Rea. Con. | | | |
| ✗ | ✗ | 86.68 | 50.00 | 24.67 |
| ✗ | ✓ | 88.33 | 50.60 | 26.97 |
| ✓ | ✗ | 88.92 | 49.33 | 25.65 |
| ✓ | ✓ | **91.44** | **53.33** | **28.95** |

Qwen2-VL (2B/7B) (Wang et al., 2024c), and InternVL-2 (8B) (Chen et al., 2024). Both models support multi-image input, enabling prompt concatenation with multi-image context. For efficiency, LVLMs with over 7B parameters are implemented in 4-bit quantization by AWQ (Lin et al., 2024a) on a single 4090 24GB GPU. Besides, we utilize the frozen BERT-base model and the ViT-L followed by a 2-layer MLP both adapted from PreFLMR (Lin et al., 2024b) as our text and vision encoders, respectively. For the setting of multiple rounds of LVLMs generation, we set $N_c = N_p = 10$, $N_s = N_m = 5$. For the setting of our MCTS-HR, we adopt the same number of few-shot samples with $K = 3$, *i.e.*, a maximum tree depth of 3. The number of initial retrieval examples is set to $N = 20$ as the action space of MCTS-HR. The maximum width of the tree is set to 3 for more action exploration. We set the default rollouts with $P = 10$, and the reward weight with default $\alpha = 0.2$. Section 4.4 details

more discussion about these parameters.

## 4.3. Main Results

Tab. 2 demonstrates the comparison results with representative methods using various LVLMs on reasoning VQA datasets, including ScienceQA, MMMU, and MathV. As in Tab. 2, Vanilla-RAG (top retrieval) (Lin et al., 2024b) has achieved a performance improvement compared to both Zero-Shot and In-Context Learning (ICL) (Han et al., 2023) with random retrieval examples on most datasets, suggesting that semantic-aware example selection is crucial for LVLMs' reasoning. In particular, our proposed RCTS demonstrates substantial gains across all benchmarks. Notably, for Qwen2-VL (2B), RCTS achieves 78.99% on ScienceQA, surpassing both Zero-Shot by +11.81% and Vanilla-RAG by +7.05%. The improvements are even more pronounced in the mathematical reasoning dataset, RCTS elevates Qwen2-VL (7B) from 24.67% (Vanilla-RAG) to 28.95%, establishing new state-of-the-art results.

Additionally, we evaluate non-reasoning VQA datasets with VizWiz and VSR-MC. Given that responses in these datasets

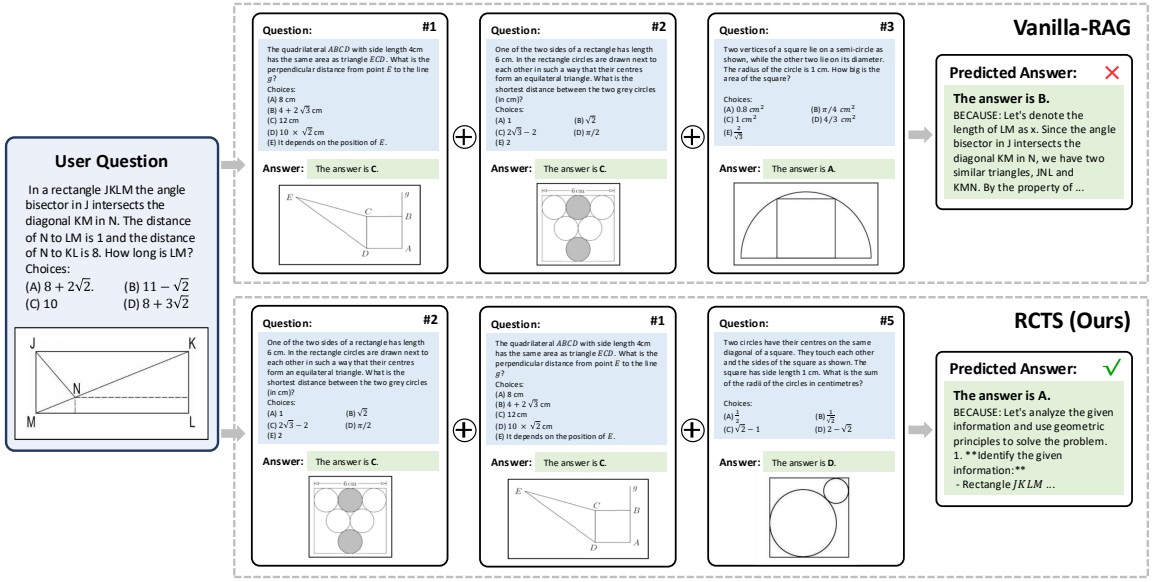

*Figure 6.* Comparison between our RCTS and the Vanilla-RAG (Lin et al., 2024b). Wherein the top examples are retrieved from the knowledge base, the below examples are re-ranked by our MCTS-HR.

*Table 5.* Ablation study of the importance weight $\alpha$ of rewards.

| $\alpha$ | ScienceQA | MMMU | Math-V |
|---|---|---|---|
| 0.0 | 90.99 | 50.67 | 26.64 |
| 0.2 | 91.44 | 53.33 | 28.95 |
| 0.5 | 90.71 | 54.00 | 27.67 |
| 0.8 | 90.17 | 54.00 | 25.32 |
| 1.0 | 86.72 | 48.67 | 25.98 |

*Table 6.* The accuracy (%) of the reasoning context on different knowledge bases.

| BaseVLM | ScienceQA | MMMU | MathV |
|---|---|---|---|
| Qwen2-VL (2B) | 90.56 | 98.60 | 85.78 |
| Qwen2-VL (7B) | 100.0 | 99.89 | 96.67 |
| InternVL-2 (8B) | 97.37 | 96.00 | 92.84 |

typically consist of a single word or a brief sentence, we only introduce the knowledge base without reasoning context. As presented in Tab. 3, our approach demonstrates consistent effectiveness with +1.61% and +3.05% enhancements on VizWiz and VSR-MC respectively compared to Vanilla-RAG, confirming its versatility and robustness.

## 4.4. Ablation Study

**Key Components.** To validate the effectiveness of key components in our RCTS, we separately eliminate the reasoning context and MCTS-HR evaluating on various VQA datasets. As shown in Tab. 4, using MCTS-HR or Reasoning Context alone always has a positive effect, such as MCTS on ScienceQA (+2.24%) and Reasoning Context on MathV (+2.3%) in Qwen2-VL (7B). The same model applies to the following. Our full method with both MCTS and reasoning context achieves better performance across all datasets, suggesting that the designed two mechanisms complement and enhance each other. Besides, the wide variety of questions covered by the MMMU results in limited performance improvement (+3.33%), attributable to an insufficient number of analogous samples within the knowledge base.

**Rewards in MCTS.** Fig. 5 (a) shows performance comparisons using self-reward alone, mutual-reward alone, and

hybrid-reward on three datasets. Obviously, using hybrid-rewards performs best on all three datasets, validating our design intent. Besides, as shown in Tab. 5, we perform a sensitivity analysis on the importance weight $\alpha$ of hybrid rewards and the default value for $\alpha$ was set to $0.2$.

**Different Rollouts.** The number of rollouts is an important factor in RCTS performance. Fig. 5 (b) shows the performance on the three datasets with different rollouts. It can be seen that as the number of rollouts increases, the performance on the three datasets shows a consistent trend. We finally set the rollouts with $P = 10$ to balance the computational overhead and performance.

## 4.5. Discussion

**Reliability of Reasoning Context.** Tab. 6 demonstrates the reliability and accuracy of the reasoning context generated by our self-consistency evaluation strategy. Specifically, we evaluate the accuracy of the ground-truth answer with the predicted answer, which is generated by splicing the question and the reasoning context into a prompt that yields the corresponding answer. As illustrated in Tab. 6, the generated reasoning context provides precise and comprehensive responses for simpler datasets like ScienceQA. For more complex questions, our strategy still yields a substantial

proportion of correct reasoning context. These results underscore the effectiveness of our proposed method.

**Qualitative Analysis.** To further illustrate our RCTS superiority over the baseline method in terms of in-context learning, we present a qualitative analysis comparing our method and Vanilla-RAG in Fig. 6. Although the Vanilla-RAG method (Lin et al., 2024b) combined with reasoning context samples can answer the reasoning information, the retrieved samples are ill-fitting, resulting in an incorrect response. In contrast, our RCTS offers more reliable reasoning contexts by re-ranking the retrieved samples and scoring all the re-ranked context sequences through our heuristic reward mechanism, providing a more reliable answer. Appendix D provides more complete re-ranking processes.

## 5. Conclusion

In this work, we introduce a multimodal RAG framework, termed RCTS, that focuses on constructing a comprehensive knowledge base with reasoning contexts and re-ranking high-quality context. The goal is to enhance the VQA ability of LVLMs by incorporating more relevant reasoning contexts, so that these models go from roughly knowing to better understanding the intrinsic knowledge of the context. Specifically, we introduce a self-consistent evaluation mechanism for generating reasoning contexts to enrich the knowledge base. In addition, MCTS-HR is proposed to re-rank the retrieved samples. Experiments on various VQA datasets show that our method is superior to in-context learning and vanilla multimodal RAG methods.

**Limitations.** Although RCTS brings significant performance improvements, it still depends on whether the presence of helpful samples is within the knowledge base. Besides, our method inevitably takes more computational overhead; the trade-off between performance improvement and model overhead is still worth discussing.

## Impact Statement

The proposed multimodal RAG framework, RCTS, significantly advances the capabilities of LVLMs in VQA by integrating a comprehensive knowledge base, reasoning contexts, and heuristic-based tree search. This innovation is pivotal for applications requiring complex multimodal VQA, such as autonomous systems, educational technologies, and AI-driven decision support systems. The framework's ability to automatically generate reasoning contexts and optimize sample selection through advanced Monte Carlo Tree Search (MCTS) ensures that AI systems can better handle intricate real-world scenarios, fostering trust and usability in AI technologies. Furthermore, the demonstrated improvements across diverse datasets, highlight the framework's versatility and potential to revolutionize fields reliant on

multimodal AI, such as healthcare, education, and urban planning. As AI continues to permeate daily life, RCTS represents a critical step toward creating more transparent, interpretable, and cognitively aligned AI systems, ultimately enhancing their societal impact and adoption.

## Acknowledgements

This research was supported by the Strategic Priority Research Program of Chinese Academy of Sciences (Grant No. XDA0370305), the National Natural Science Foundations of China (Grant No. 62306310).

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

## A. Monte Carlo Tree Search with Heuristic Rewards (MCTS-HR)

We detail the complete workflow of MCTS below.

- **Tree Initialization:** A root node is initialized using a native user's query without any retrieved samples, generating a zero-shot response for the early stopping strategy.

- **Node Expansion:** The algorithm employs a value $Q(a)$ and the visits times $N(a)$ to rank all nodes that have not been fully expanded. The node $a$ with the highest value is selected for further exploration using a greedy sampling strategy.

- **Action Selection:** During node expansion, the MCTS employs an action sampling function $F_{\mathcal{A}}$ to sample from the action space $\mathcal{A}$, which serves as the expansion node. The action space is constructed using $N$ samples retrieved from the knowledge base.

- **Branch Simulation:** When the maximum depth is reached, the algorithm performs a simulation, often termed "rollouts". This involves concatenating all the actions along the branch with the user query to form a $K$-shot prompt, which is then used to generate the response for the branch.

- **Reward Evaluation:** The $K$-shot response is evaluated using a reward function $\mathcal{R}$ to obtain a reward value $Q$. This process incorporates self-reward feedback and answers heuristic feedback constraints, via in-context consistency to ensure reliability and fairness in scoring.

- **Backpropagation:** The reward value $Q$ of the $K$-shot response is propagated backward to its parent node and other related nodes to update the tree's value information. If the $Q$ value of any child node changes, the parent node's $Q$ is also updated accordingly.

- **UCT Update:** After updating the $Q$ values of all nodes, a collection C of candidate nodes is first identified for further expansion and selection, then use the Upper Confidence Bound for Trees (UCT) update formula to update the UCT values of all nodes for the next stage of exploration following (Silver et al., 2016; Zhang et al., 2024a). Formally, for a node $a$ that have not been fully explored, the $\text{UCT}_a$ is defined as:

$$\text{UCT}_a = Q(a) + c\sqrt{\frac{\ln N(\text{Father}(a)) + 1}{N(a) + \epsilon}}, \tag{11}$$

where $Q(a)$ is the reward value of node $a$, $N(\cdot)$ is the total visit times of given nodes, $c$ is a constant to balancing exploitation and exploration, $\epsilon$ is a small constant for avoid devided-by-zero.

The algorithm iterates through these stages until a termination condition $T$ is met, including maximum rollout constraints or reaching the early stopping strategy, continuously re-ranking the retrieved samples and improve the quality of answers, and exploring new possibilities. The termination function criteria $T$ can derive from several conditions:

- **Early Stopping:** Termination occurs when the answers of the root node and the leaf nodes, based on greedy retrieval and initial branching, are consistent.

- **Expansion Constraints:** The search terminates once the number of rollouts reaches a predetermined limit or all possible combinations of re-ranking samples have been traversed.

## B. Dataset Details

### B.1. ScienceQA

Science Question Answering (ScienceQA) (Lu et al., 2022) is a benchmark comprising 21,208 multimodal multiple-choice questions drawn from elementary and high school science curricula. As shown in Fig. 7, this dataset is enriched with detailed annotations, including correct answers, corresponding lectures, and comprehensive explanations. The questions span a diverse array of topics across three primary subjects: natural science, social science, and language science. The task involves selecting the correct answer from the provided multiple-choice options.

In our experiments, we adhere to the original dataset split, utilizing the training and validation sets, which consist of 16,967 examples, as our knowledge base. The test set, containing 4,241 examples, is employed for evaluation purposes.

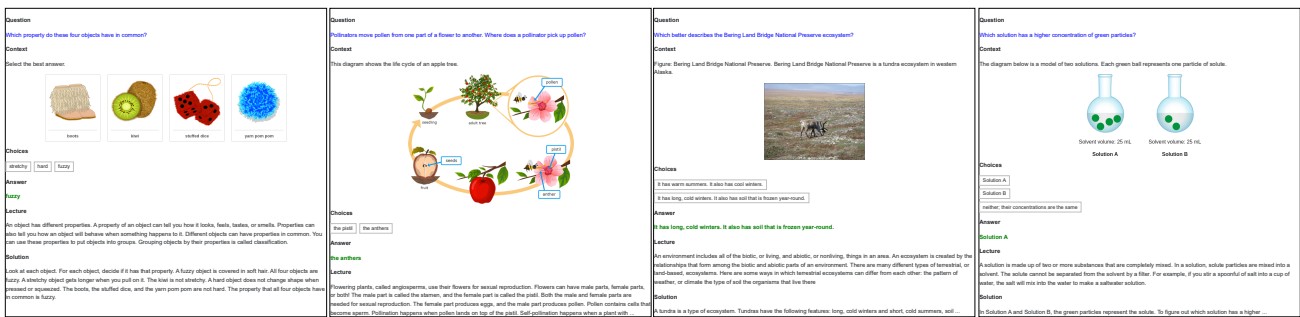

*Figure 7.* Illustrative examples from the ScienceQA dataset (Lu et al., 2022).

## B.2. MMMU

The Massive Multi-discipline Multimodal Understanding and Reasoning Benchmark (MMMU) (Yue et al., 2024) is a novel benchmark that comprises 11,550 carefully selected multimodal questions. These questions are divided into 150 for development, 900 for validation, and 10,500 for testing. This dataset is drawn from college exams, quizzes, and textbooks spanning six common disciplines: Art & Design, Business, Science, Health & Medicine, Humanities & Social Science, and Tech & Engineering. This dataset focuses on advanced perception and reasoning with domain-specific knowledge, challenging models to perform tasks similar to those faced by experts. Besides, as illustrated in Fig. 8, the questions cover a diverse array of topics across 30 subjects and 183 subfields, including 30 highly heterogeneous image types such as charts, diagrams, maps, tables, music sheets, and chemical structures. The task mainly involves selecting the correct answer from the provided multiple-choice options.

Due to the invisibility of the true samples in the test set and the broad domain coverage of the dataset, which results in low similarity between different samples, we utilize the validation set consisting of 900 samples to construct the knowledge base and the development set with 150 examples for evaluation in our experiments.

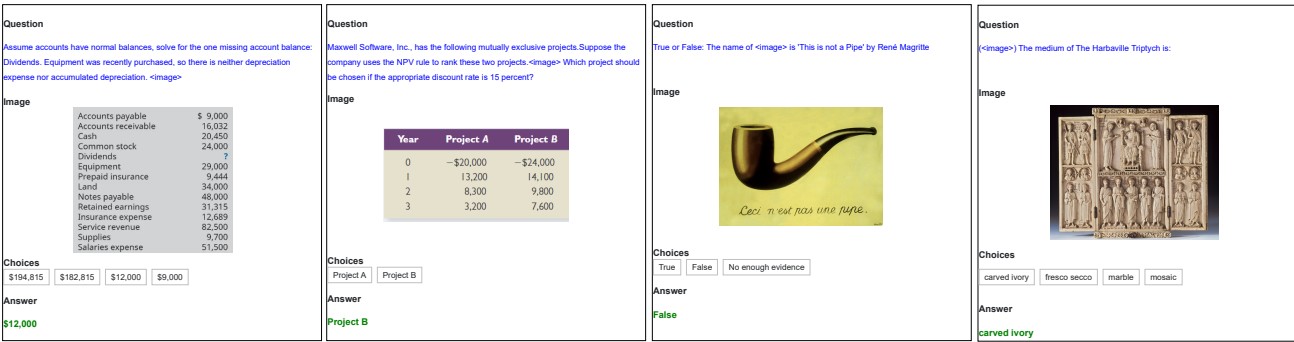

*Figure 8.* Illustrative examples from the MMMU dataset (Yue et al., 2024).

## B.3. MathV

MATH-Vision (Math-V) (Qi et al., 2024) is a benchmark designed to evaluate the multimodal mathematical reasoning capabilities of foundation models across a wide range of mathematical tasks with visual contexts. It comprises a total of 3040 multimodal math questions, covering 16 subjects, including Algebra, Analytic Geometry, Arithmetic, Combinatorial Geometry, Combinatorics, Counting, Descriptive Geometry, Graph Theory, Logic, Metric Geometry, Solid Geometry, Statistics, Topology, and Transformation Geometry. As depicted in Fig. 9, this dataset spans five levels of difficulty. The task involves selecting the correct answer from the provided multiple-choice options and outputting the calculated answer straightforwardly.

In our experiments, to ensure the reliability of the knowledge base, we employ a deduplicated test set as the knowledge base,

which contains 2736 samples. The test-mini set which contains 304 samples, is employed for evaluation purposes.

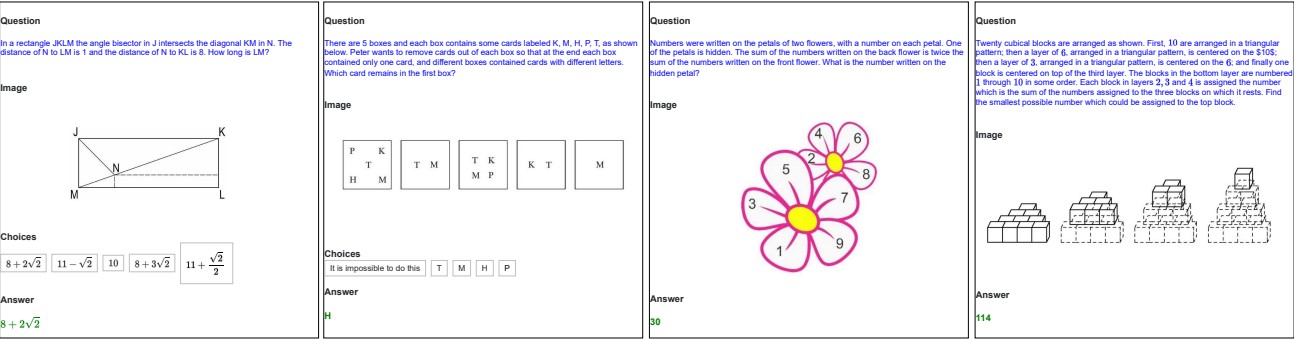

Figure 9. Illustrative examples from the MathV dataset (Qi et al., 2024).

## B.4. VizWiz

VizWiz (Gurari et al., 2018) is a Visual Question Answering (VQA) dataset designed to assist individuals with visual impairments in better understand visual information in their daily lives. This dataset comprises visual questions from blind individuals seeking answers to everyday visual inquiries. It includes a total of 20,523 training samples and 4,319 validation samples. The task in VizWiz involves determining "True" or "False" based on the provided questions and generating a concise phrase to answer each question directly. In our experiments, we utilize the training set as the knowledge base and the validation set for evaluation purposes.

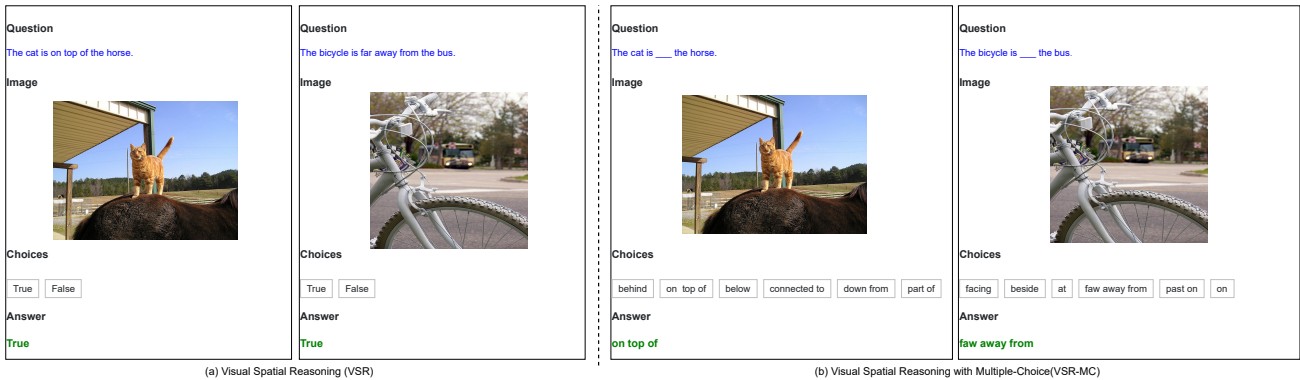

Figure 10. Illustrative examples from the Visual Spatial Reasoning dataset (VSR) (Liu et al., 2023) and Visual Spatial Reasoning with Multiple-Choice dataset (VSR-MC).

## B.5. VSR-MC

Visual Spatial Reasoning (VSR) (Liu et al., 2023) serves as a benchmark that encompasses over 10k natural image-caption pairs, featuring 66 types of spatial relations in English, such as "under", "in front of", and "facing". The primary objective of the VSR task is to assess whether the captions accurately reflect the spatial relations depicted in the images by answering True or False, while obstacle the in-context learning.

For this issue, as illustrated in Fig. 10 (b), we developed a visual spatial reasoning dataset with a multiple-choice format (VSR-MC). Specifically, for each sample instance, we masked the initially correct spatial relationship and then randomly selected five relations from the remaining 65 spatial relations to serve as candidate options, alongside the true spatial relation that functions as the question. The original correct spatial relation is designated as the standard answer.

In our experiments, we applied this pipeline to the training and validation sets of the VSR dataset, comprising 4,440 samples, to construct a comprehensive knowledge base. Similarly, we processed the test set to derive 1,181 test samples for

performance evaluation. This data processing strategy ensures the rigor and comparability of the experimental results.

# C. Prompts in Experiment

## C.1. Reasoning Context

This section provide the detailed prompt for Section 3.2.

**Get Reasoning Context:**

SYSTEM: You are a helpful assistant tasked with providing a detailed and structured thought process based on the answer. The thought process should be logically sound, step-by-step, and clearly lead to the final answer.

USER: **User Question:**
{Question}
**Answer:**
{Answer}
**System Prompt:**
Please describe your thought process in a step-by-step, structured manner, ensuring that each step logically leads to the final answer. Let's think step by step.

**Get Predicted Answer by Reasoning Context:**

SYSTEM: You are a helpful assistant.

USER: **User Question:**
{Question}
**THOUGHT PROCESS:**
{Thought Process}

## C.2. Answer Prediction

This section provides the detailed prompt for our experiments. Our prompts for different datasets are primarily adapted from VLMEvalKit (Duan et al., 2024), with necessary modifications to ensure compatibility across the specific tasks and datasets under investigation.

**Zero-Shot & Few-Shot** (ScienceQA)

SYSTEM: You are a helpful assistant. When given a question and an image, please analyze the content and provide your answer in the specified format below:
''' The answer is X. BECAUSE: [Your detailed reasoning] '''
- X must be one of the options.
- [Your detailed reasoning] should clearly explain the rationale behind your choice.
**Important:** Adhere strictly to the above format without deviations.

USER: {Question}

ASSISTANT: {Answer}

...

USER: {Question}

**Few-shot-with-reasoning-context** (ScienceQA)

SYSTEM: You are a helpful assistant responding to the question according to the context. When given a question and an image, please analyze the content and provide your answer in the specified format below:
'''
**THOUGHT PROCESS:**
[Your thought process for arriving at the answer].
**FINAL ANSWER:**
The answer is X.
BECAUSE: [Your detailed reasoning].
'''
- [Your thought process for arriving at the answer] should provide a step-by-step process that led to your chosen answer.
- X must be one of the options: A, B, C, D, E.
- [Your detailed reasoning] should clearly explain the rationale behind your choice.
**Important:** Adhere strictly to the above format without deviations.

USER: {Question}

ASSISTANT: {Answer}

...

> USER: {Question}

## Zero-Shot & Few-Shot (MMMU)

> SYSTEM: You are a helpful assistant.

> USER: {Question}
> Answer with the option letter from the given choices in the following format: 'The answer is X.' (without quotes) where X must be one of options.

> ASSISTANT: {Answer}

...

> USER: {Question}
> Answer with the option letter from the given choices in the following format: 'The answer is X.' (without quotes) where X must be one of options.

## Few-shot-with-reasoning-context (MMMU)

> SYSTEM: You are a helpful assistant.

> USER: {Question}
> Answer with the option letter from the given choices in the following format: 'The answer is X. BECAUSE: xxx' (without quotes) where X must be one of options. Think step by step before answering.

> ASSISTANT: {Answer}

...

> USER: {Question}
> Answer with the option letter from the given choices in the following format: 'The answer is X. BECAUSE: xxx' (without quotes) where X must be one of options. . Think step by step before answering.

## Zero-Shot & Few-Shot (MathV)

> SYSTEM: You are a helpful assistant.

> USER: {Question}
> Answer the preceding multiple choice question. The format of your response should follow this format: 'The answer is //boxed{X} or //boxed{YOUR_ANSWER}.' (without quotes), where 'X' must be one of the options or 'YOUR_ANSWER' is your conclusion.

> ASSISTANT: {Answer}

...

> USER: {Question}
> Answer the preceding multiple choice question. The format of your response should follow this format: 'The answer is //boxed{X} or //boxed{YOUR_ANSWER}.' (without quotes), where 'X' must be one of the options or 'YOUR_ANSWER' is your conclusion.

## Few-shot-with-reasoning-context (MathV)

> SYSTEM: You are a helpful assistant.

> USER: {Question}
> Answer the preceding multiple choice question. The format of your response should follow this format: 'The answer is //boxed{X} or //boxed{YOUR_ANSWER}. BECAUSE: xxx' (without quotes), where 'X' must be one of the options or 'YOUR_ANSWER' is your conclusion. Think step by step before answering.

> ASSISTANT: {Answer}

...

> USER: {Question}
> Answer the preceding multiple choice question. The format of your response should follow this format: 'The answer is //boxed{X} or //boxed{YOUR_ANSWER}. BECAUSE: xxx' (without quotes), where 'X' must be one of the options or 'YOUR_ANSWER' is your conclusion. Think step by step before answering.

# D. Case Example of RCTS

In this section, we primarily illustrate the re-ranking process of our proposed Monte Carlo Tree Search with Hybrid Re-ranking (MCTS-HR) framework. As illustrated in the experimental analysis, we present comparative visualizations spanning mathematical reasoning (Fig. 11 and Fig. 12), chart interpretation tasks (Fig. 14 and Fig. 13), and natural scene image comprehension (Fig. 15 and Fig. 16).

Besides, Fig. 15 and Fig. 16 exemplify two distinctive scenarios. Fig. 15 represents an ideal case where near-identical reference sample exist in the knowledge base, enabling the Vanilla-RAG to directly retrieve the matching sample and consequently ensure all candidate branches yield correct answers. Conversely, Fig. 16 demonstrates a challenging scenario where no semantically similar samples are available in the knowledge base, resulting in erroneous outputs across all candidate branches due to excessive dissimilarity between existing references and the query instance. This scenario highlights the critical dependency of retrieval performance on the knowledge base's coverage and semantic granularity.

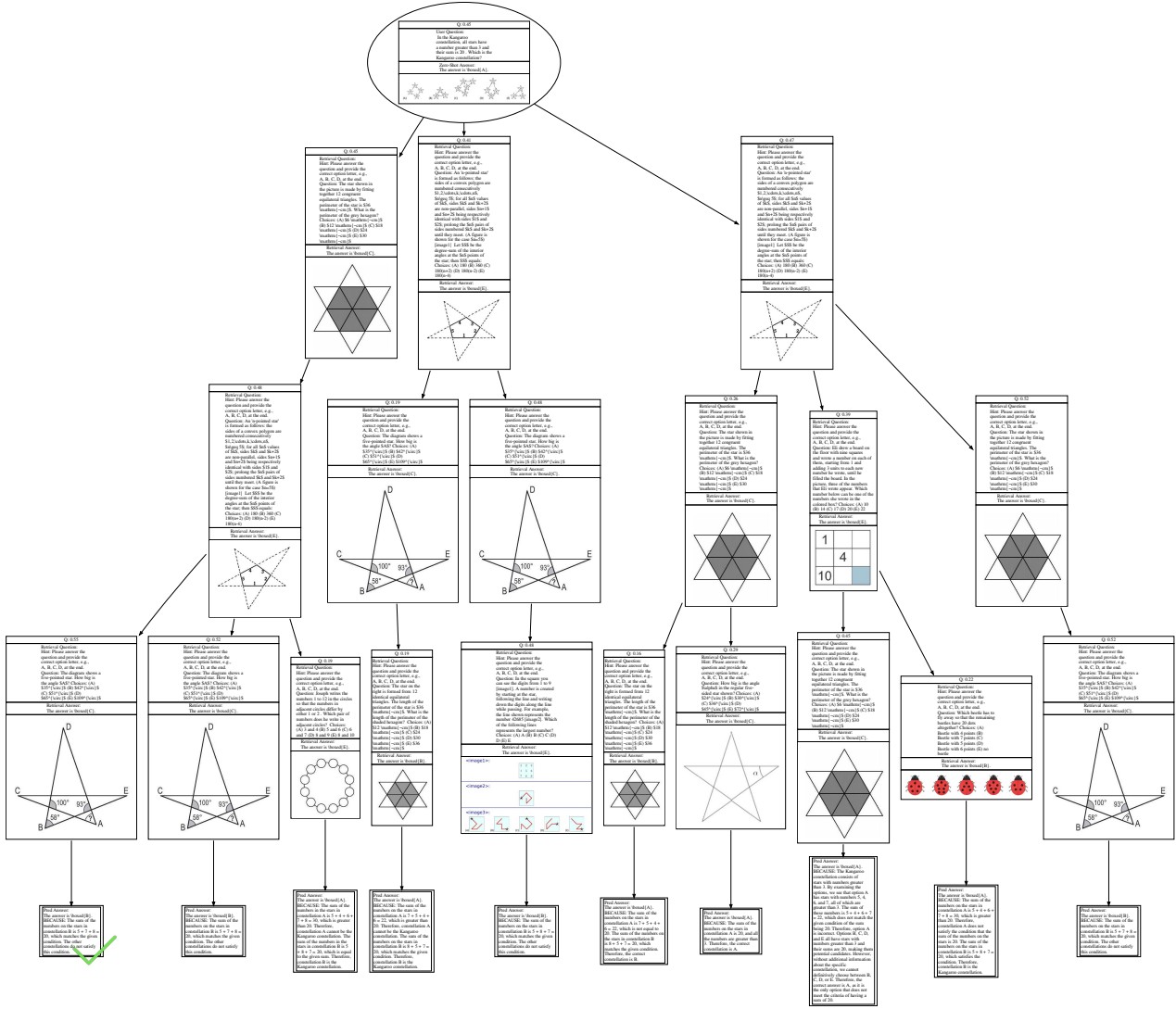

*Figure 11.* Illustration of the MCTS re-ranking process on math question.

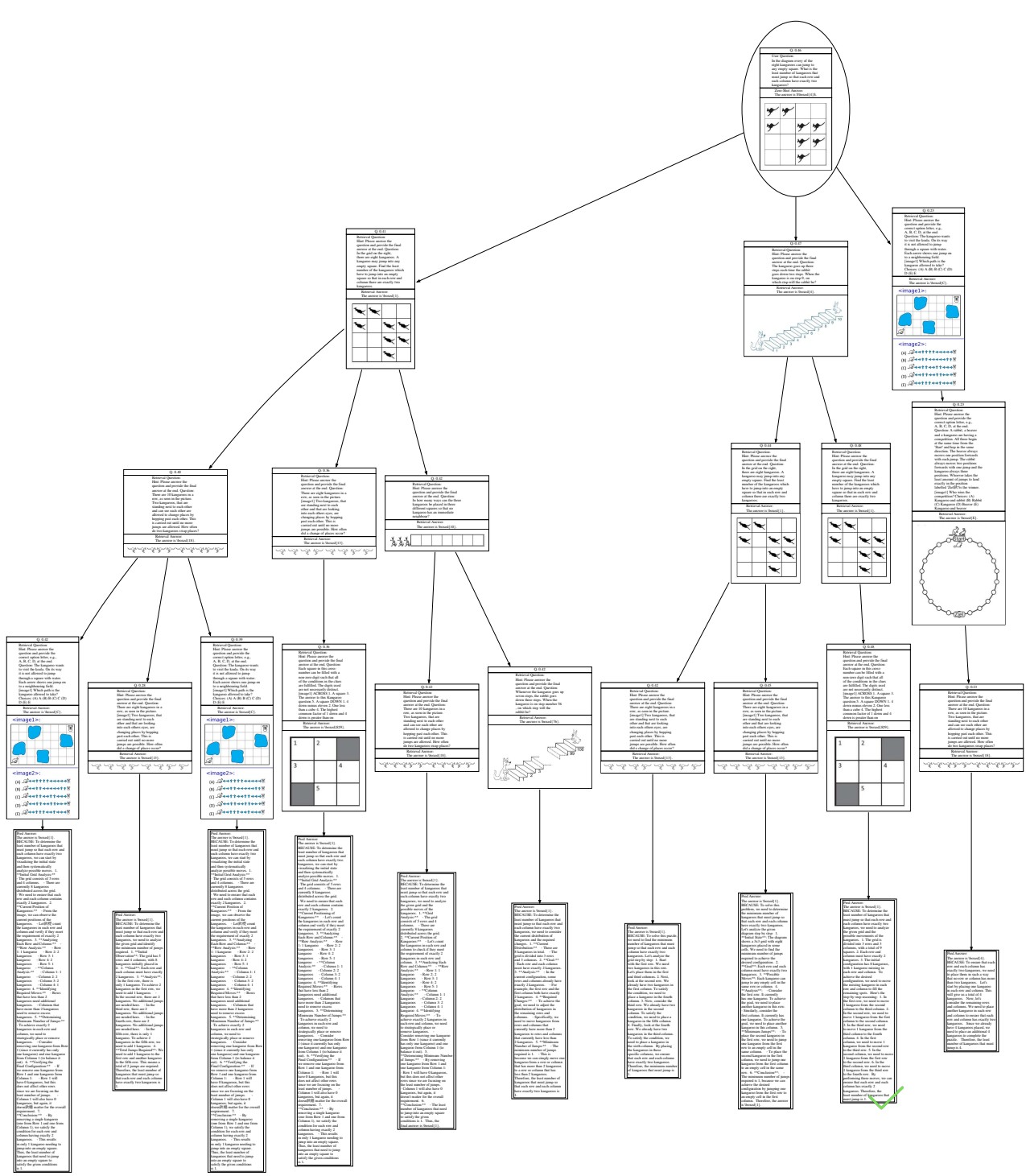

*Figure 12.* Illustration of the MCTS re-ranking process on math question.

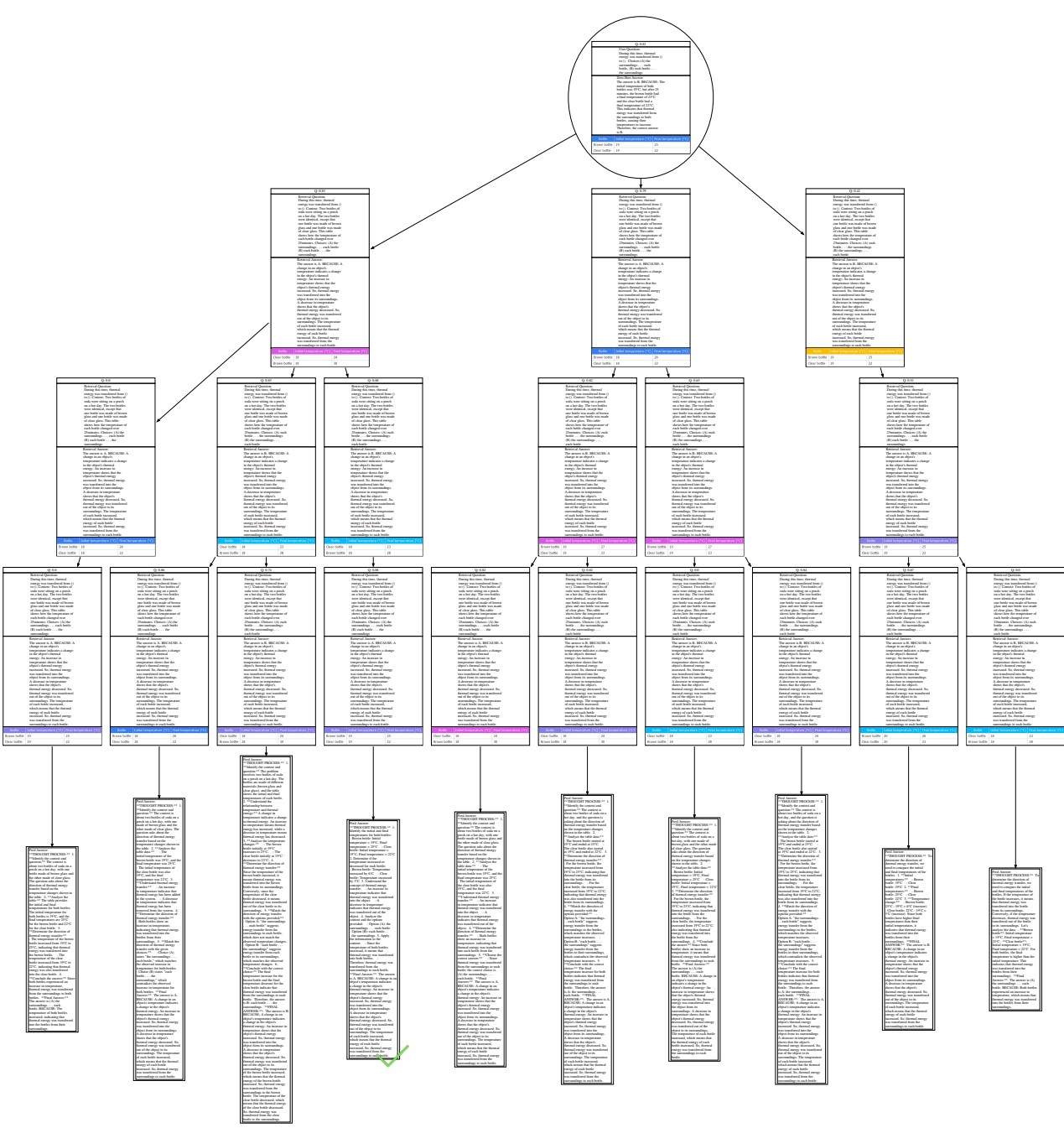

*Figure 13.* Illustration of the MCTS re-ranking process on table question.

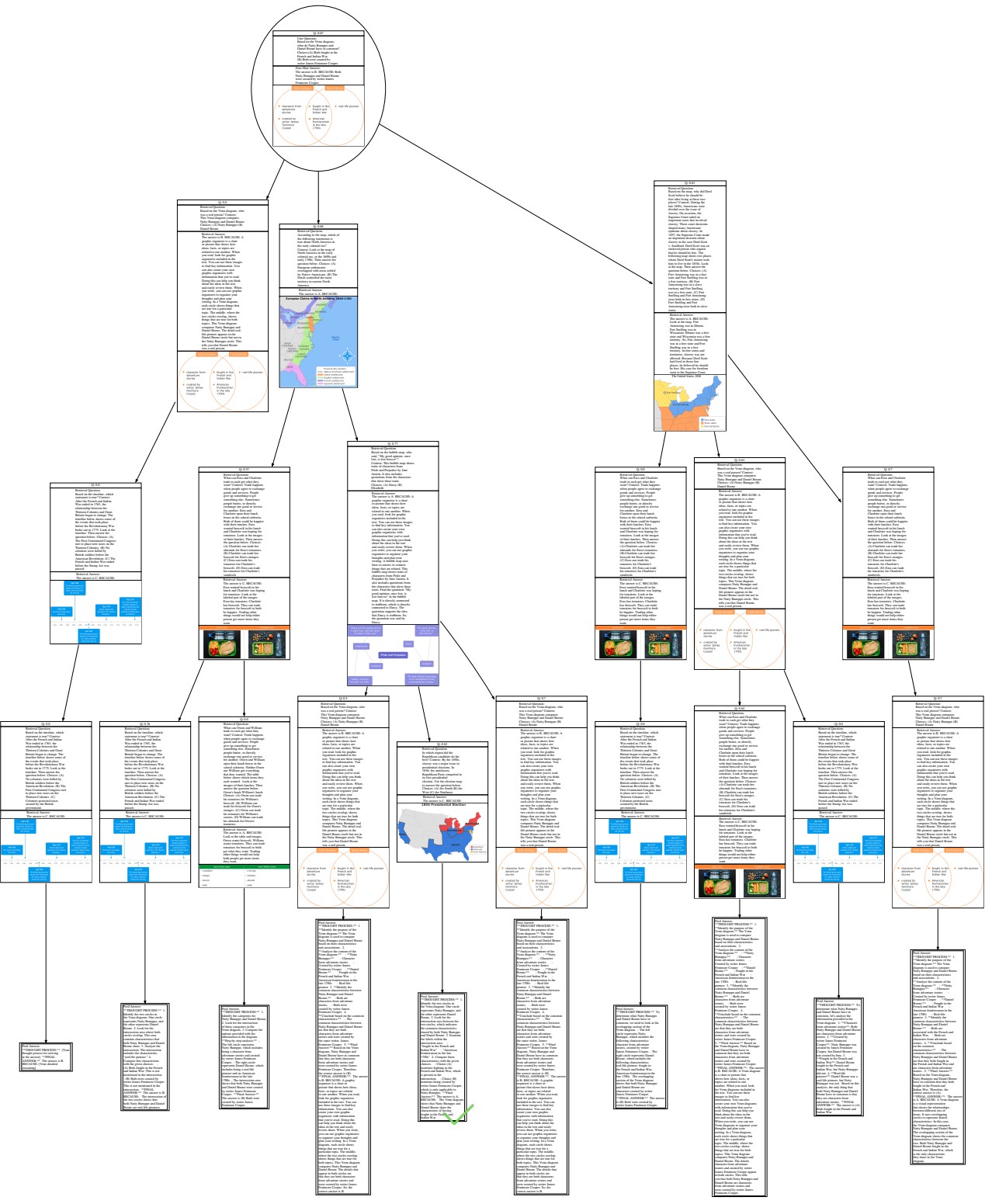

*Figure 14.* Illustration of the MCTS re-ranking process on chart question.

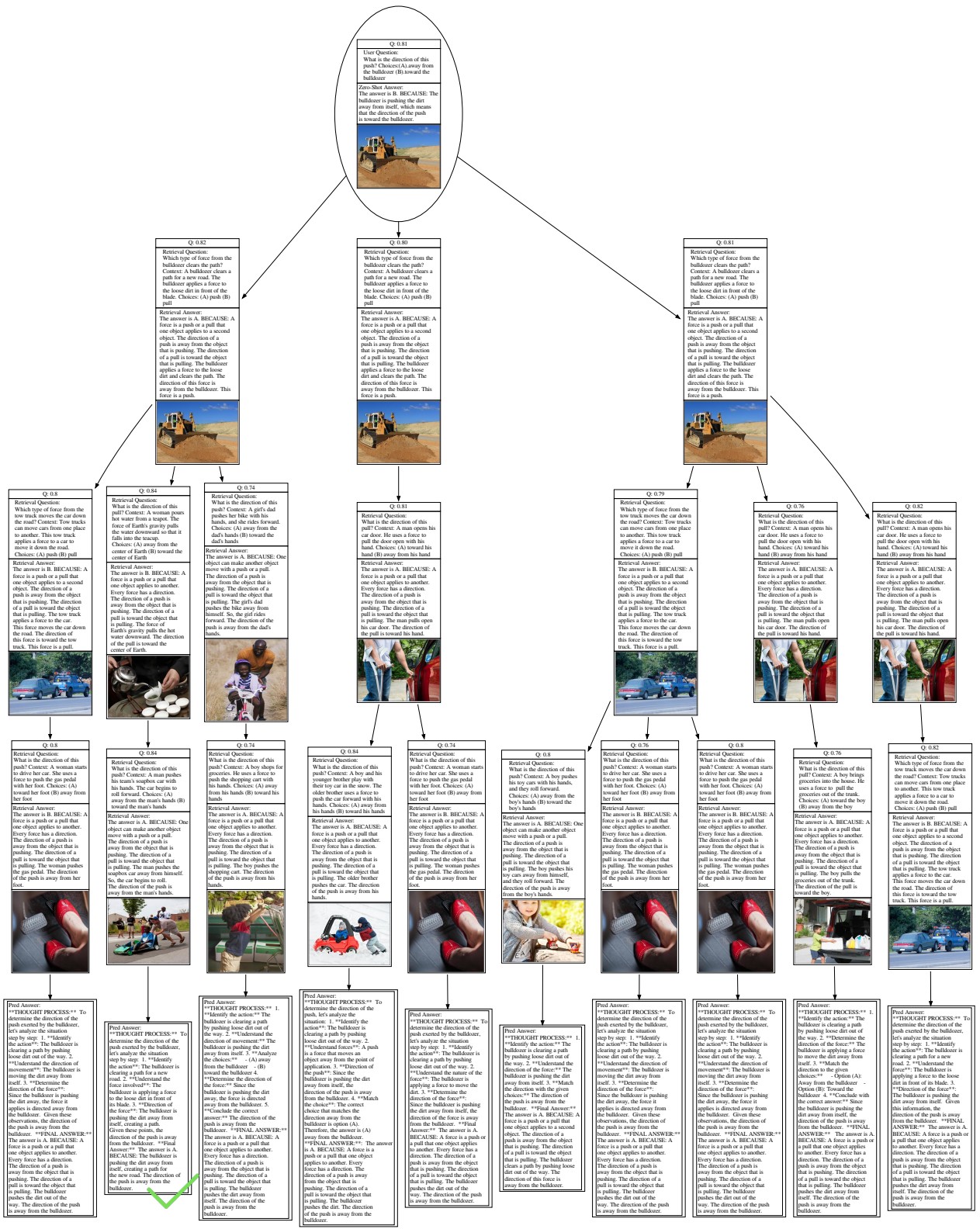

*Figure 15.* Illustration of the special case of the MCTS re-ranking process on natural question.

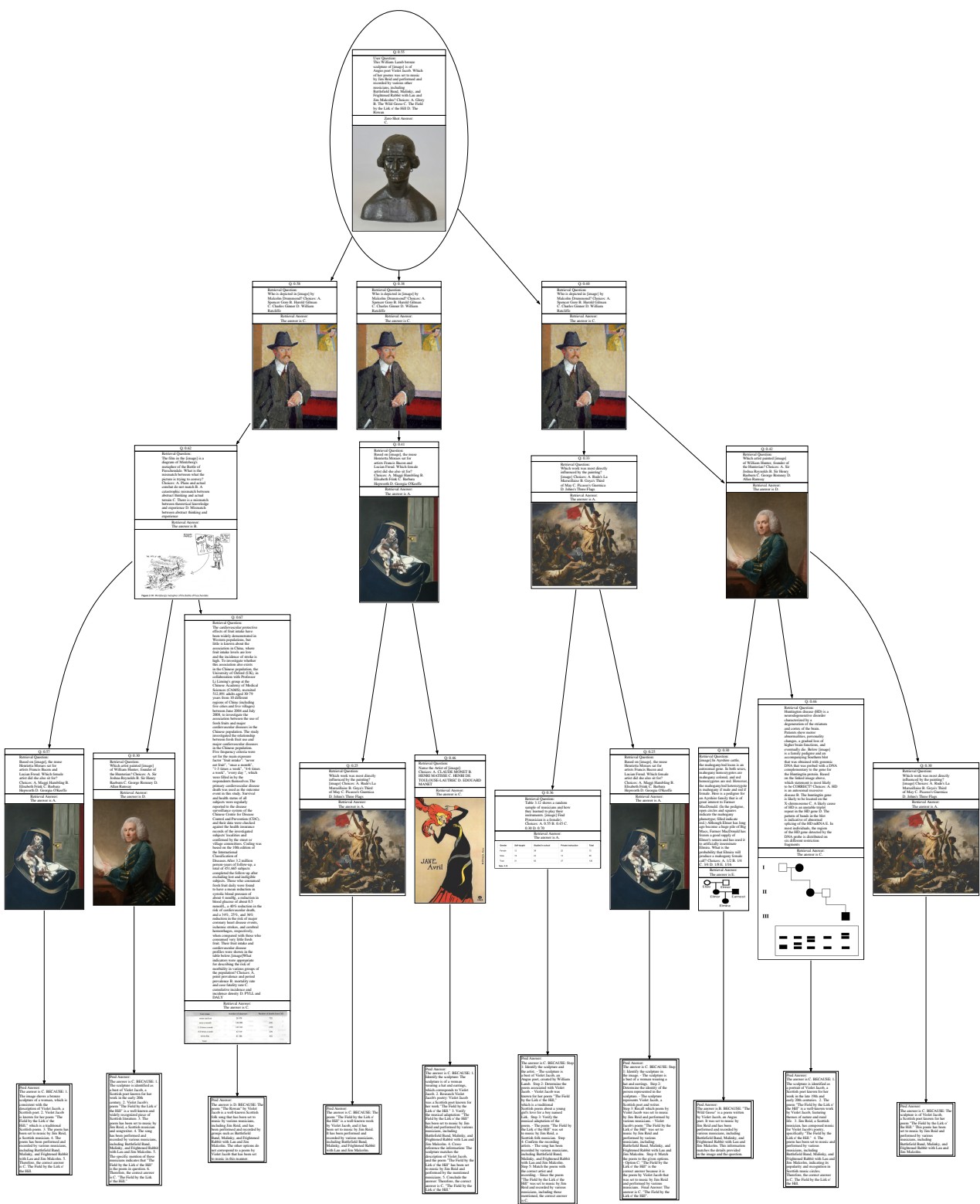

*Figure 16.* Illustration of failure case of the MCTS re-ranking process on art question.

