# OpenReview forum: "Re-ranking Reasoning Context with Tree Search Makes Large Vision-Language Models Stronger"
_ICML.cc/2025/Conference — ICML 2025 spotlightposter_

### Official Review · Reviewer_yd6M · 2025-03-09

**Overall Recommendation:** 3

**Summary:**

This paper introduces a novel framework to enhance Large Vision Language models (LVLMs) capability for visual question answering. Two major contributions are demonstrated in this paper. First is about creating a  comprehensive knowledge base enriched with automatically generated reasoning contexts. Second is about employing a tree search-based re-ranking mechanism called MCTS-HR, which strategically orders retrieved examples to improve the accuracy of the LVLMs' responses.

The experiments demonstrate that RCTS outperforms other methods on various visual reasoning datasets by better guiding the LVLMs to understand and utilize contextual information. Further, ablation studies on the proposed methods also validate the contributions of its key components.

**Claims And Evidence:**

From the motivation, method and experimental results of this paper, there are three major claims being made:
1. The proposed RCTS achieves state-of-the-art performance acorss the board: The author shows how the proposed method performs on multiple reasoning VQA benchmarks, like ScienceQA, MMMU, MathV, etc. For example, on ScienceQA, RCTS achieved a 78.99% accuracy with Qwen2-VL (2B), surpassing Zero-Shot by +11.81% and Vanilla-RAG by +7.05%.
However, from the major Table 2, it is not clear why for InternVL-2 method, the Vanilla-Rag performs worse than the Zero-shot on three benchmarks. Is that because of the quantization method for models over 7B? The author needs to add explanations on this.
2. MCTS-HR effectively re-ranks retrieved examples: By re-ranking the retrieved samples, RCTS ensures that LVLMs leverage high-quality contextual reasoning. In Figure 6, the author shows the comparison between Vanilla RAG and RCTS.
3. Hybrid rewards in MCTS-HR are beneficial: Figure 5 shows that using hybrid rewards in MCTS-HR leads to the best performance compared to using either self-reward or mutual-reward alone.

**Essential References Not Discussed:**

Not aware of important missings in literature.

**Experimental Designs Or Analyses:**

As I stated in the Claims And Evidence section, my concern is majorly about why, in Table 2 for InternVL-2 method, the Vanilla-Rag performs worse than the Zero-shot on three benchmarks? I could not find the discussion about this in the paper. And does this finding consistant with the other RAG papers using InternVL-2?

**Methods And Evaluation Criteria:**

Methods:

The overall methods are reasonable. One concern after reading the paper, which is also missed in the experiments, is about the RAG cost. I understand the knowledge base could be built beforehand. But it is also important to show the database building cost, retrieving cost, and the reranking cost to help the readers better understand the proposed method.

Criteria:

The paper follows standard evaluation criteria with other LVLMs and RAG methods, which makes sense to the problem. Also, the author includes both reasoning and non-reasoning VQA benchmarks to show the advancement of the proposed method.

**Other Comments Or Suggestions:**

No, I do not have other comments.

**Other Strengths And Weaknesses:**

1. One thing not clear is about the retrieving key of the method. The embedding of the query image and text might not have very high similarity with correlated information in the database, especially when the query requires long reasoning chains. For example, the information in the database is not word-similar to the query question but might be an important thinking step to respond to the query. It is not clear how this has been considered in the paper.

2. I could not find the discussion of the option of N in the experiments. Will the method still work when N is very small, and will it bring marginal benefits when increasing it?

**Questions For Authors:**

No, I do not have other questions for the authors.

**Relation To Broader Scientific Literature:**

The paper is related to LVLMs, multimodal information retrieval and RAG. These fields have been discussed in the related works.

**Theoretical Claims:**

There is no theoretical claims in this paper.

---

> ### Author Rebuttal · Authors · 2025-04-01
>
> Thanks for your constructive reviews and address your concerns as follows.
>
> **Q1**: From the Table 2, it is not clear why for InternVL-2, the Vanilla-Rag performs worse than the Zero-shot. Is that because of the quantization method for models over 7B?
> **A1**: Thank you for pointing out this issue. Actually, we initially had similar concerns and conducted multiple experiments, obtaining consistent results. Here are some possible reasons we analyzed:
> 1. This issue may not be caused by model quantization, since we use the same quantized models for all experiments. The performance degradation is consistent across all methods.
> 2. Since InternVL-2 has been trained on ScienceQA and MathQA datasets, this might be because ICL alters the distribution of the model's outputs, resulting in failed predictions.
> 3. Compared to Qwen2-VL, we observe that Qwen2-VL has better performance in multiple rounds of interaction capabilities, while our RCTS implemented ICL through multi-turn conversations. This may be the reason why vanilla-RAG is inferior to zero-shot in InternVL-2.
> 4. As for why RCTS did not encounter this issue, this might be attributed to the fact that MCTS-HR incorporates heuristic rewards, which helps to validate the predicted results and can therefore mitigate the impact of ICL on the model distribution to a certain extent.
>
> **Q2**: It is important to show the database building cost, retrieving cost, and the reranking cost to help the readers better understand.
> **A2**: Regarding the cost issues of our model, all cost experiments are evaluated on an A800 GPU, using vllm 0.6.4, with Qwen2-VL-7B-GPTQ-Int4. Additionally, we randomly select 100 samples from ScienceQA and MathV datasets for cost comparison, and repeat tests 5 times to mitigate hardware fluctuations.
> 1. KB Construction Cost. The time required to construct the KB is positively correlated with the difficulty of the problems. This is because once the Score (L202) reaches the maximum value, the current reasoning context is returned as the optimal reasoning path. The cost-time is shown below:
> |Cost-Time (seconds)|ScienceQA|MathV|
> |-|-|-|
> |KB Construction (per)|0.94±0.06|4.28±0.22|
>
> 2. Hybrid Retrieval Cost. As mentioned in Q4 by Reviewer wbuQ, we employ Bert-Base + ViT-L (422M parameters) to extract text and image embeddings. Besides, we pre-store the features from KB using Faiss for fast retrieval. The average retrieval time per sample ranges from 5–30 ms, which is significantly shorter than the inference time required for MCTS we discuss below.
> 3. MCTS Inference Cost. MCTS inherently requires more simulations due to its rollout mechanism, thus consuming more cost. To enable faster responses for simpler questions, we introduced an early-stopping strategy based on answer consistency (L642–645), i.e., if the initial branch and the greedy retrieval branch yield the same answer, the result is returned, bypassing MCTS’s multi-round simulations. Therefore, our inference cost also varies depending on the difficulty of the problem. The cost time of MCTS-Reranking is shown as below:
> |Cost-Time (seconds)|ScienceQA|MathV|
> |-|-|-|
> |MCTS-Reranking (per)|29.55±4.5|62.32±8.6|
>
> **Q3**: One thing not clear is about the retrieving key of the method.
> **A3**: Thank you for pointing this out. Actually, we have observed this issue in L60 under the "challenge ii)". Specifically, our hybrid retrieval method computes text embeddings using the user's question and the KB questions, without including the long reasoning contexts and corresponding answers (L187-191, L211-213). This design ensures efficient retrieval of the most similar image-text pairs from the KB that align with the user’s query.
> However, relying solely on feature similarity is insufficient, as the information in the KB which is not similar to the query might be an important thinking step to respond to the query. For this issue, we propose a tree search approach with Heuristic Rewards, termed MCTS-HR, which dynamically re-ranks and selects the most related samples rather than similar samples. By evaluating candidate samples through heuristic rewards, MCTS-HR identifies the most pertinent samples (i.e., those beneficial for addressing the user's question) from the candidate feature-similar samples (L209-211).
>
> **Q4**: The discussion of the option of N in the experiments.
> **A4**: Thank you for pointing this out. We take more ablation studies on N with Qwen2VL-7B, conducted on ScienceQA and MathV dataset.
> It can be observed that a smaller N degrades performance, since the reduction in candidate similarity samples narrows the action space of MCTS, limiting its ability for re-ranking. On the other hand, selecting an excessive number of N introduces more noise into the MCTS action space, i.e., samples that are neither similar nor particularly helpful. Taking these into account, we set N to 20.
> |N|ScienceQA|MathV|
> |-|-|-|
> |5|89.4|26.0|
> |10|90.6|27.0|
> |15|90.7|27.6|
> |20|91.4|29.0|
> |30|91.2|26.3|
> |50|91.3|27.6|

---

### Official Review · Reviewer_wbuQ · 2025-03-11

**Overall Recommendation:** 3

**Summary:**

This paper focuses on RAG for VQA tasks. The authors propose constructing a reasoning-based KB with examples of successful reasoning and then introduce an MCTS-based method for finding the best set of ICL examples, motivated by the fact that existing models can only take a fairly small number of ICL examples (compared to what is retrieved).

The approach first constructs a KB of reasoning that leads to correct outputs. The main contribution is the MCTS search method, which helps choose which example to select by using the consistency of the answer (were an example to be selected) as heuristic reward, combined with a mutual consistency reward. Here, the authors formulate retrieval as a sequential decision-making task, where the action space consists of selecting examples from the KB. They start by retrieving a set of relevant examples with vector similarity and then rerank according to their heuristic tree-search algorithm.

The approach is evaluated on multiple datasets: ScienceQA, MMMU, MathV, VizWiz, VSR-MC and across two strong VLMs. The results unequivocally demonstrate improvements over examples retrieved with vanilla RAG and randomly-retrieved examples.

The authors ablate key parts of their method, including their rewards, showing that both rewards are required for strong performance.

## update after rebuttal
The rebuttal has clarified some of the questions I had. I will maintain my positive recommendation.

**Claims And Evidence:**

- the claims are clear: the authors are claiming that by selecting examples for ICL via their method they can make better use of their generated KB
- the evidence largely points to this, however, I think the setup for the Vanilla-RAG baseline is not completely clear. It's later improved by Figure 6 but this information comes too late and it is still not completely clear to me what the source of the examples used in Vanilla RAG is.

**Essential References Not Discussed:**

- https://arxiv.org/pdf/2307.07164 uses a reward model to learn to retrieve ICL examples
- https://arxiv.org/pdf/2402.07812 seems to also use MCTS combined with retrieval for guiding model outputs, along with a proxy reward.

**Experimental Designs Or Analyses:**

Experimental design is sound.

**Methods And Evaluation Criteria:**

- The evaluation domains make sense, and the method is evaluated on a range of datasets and across two recent models.
- The proposed method performs strongly but the description of the method is not very clear. There are a few parts that were unclear to me:
1. L200-201: what is the score being used here? Why do you need SC if you have ground-truth answers?
2. L256-259: the purpose of these rewards (especially the mutual consistency reward) should be made clear earlier on.
3. How are images/text embedded? Which model is used?
4. Which model is used to construct the KB?

Overall, the methods section could benefit from more sign-posting on why particular decisions are being made. Right now it reads like a sequence of somewhat arbitrary decisions; the ablations later show that these decisions help but there is little intuition given on why they should help.

**Other Comments Or Suggestions:**

I have a few small quibbles about the language in the paper:
- The "known" vs "understood" claim is not clear and in my opinion doesn't add much to the introduction, the authors should be clearer on what they mean by understanding, which is a pretty hazy concept
- L162 "humans always learn by examples": this is a big claim and not at all settled fact, it either needs more evidence or should be hedged.

Typos:

- L211: "solely on single-modal." -> "solely on a single modality"
- typo in fig 6 caption (retrievd)
- L609: Native, retrieved
- L630: Formally
- L736: backwards quotes (also in the rest of appendix)

**Other Strengths And Weaknesses:**

Strengths:
- overall the results of the paper are strong, with consistent gains across domains
- the idea of using self-consistency as a signal for MCTS this way is widely applicable

Weaknesses:
- It's not very clear what the action space is or why this needs to be a sequential problem. In this case, the authors are selecting a set of ICL examples. It doesn't seem to me like order matters here, i.e. does selecting a different example first makes a difference? If not, why is it being modeled as a sequential task? An ablation here on how much order matters would be helpful.
- Computational cost is mentioned in the limitations, it would be good to have a number to put to this (as the MCTS method seems like it could involve a high number of calls)

**Questions For Authors:**

- What dataset is Vanilla RAG retrieving from? Is the info coming from the same KB but without MCTS reranking?
- L262 "employs QA pairs retrieved... as candidate actions", does this mean that you are treating selecting a QA pair is an action?
- (from methods:) How are you embedding images and text? Which model is used?
- Which model is used to construct the KB?

**Relation To Broader Scientific Literature:**

The contribution of the paper relates more broadly to the question of data selection under a budget. The idea of using MCTS for ICL selection in this way seems novel and could be applied to other domains as well. In general, picking the right examples from a superset is an important problem.

**Theoretical Claims:**

No theoretical claims made.

---

> ### Author Rebuttal · Authors · 2025-04-01
>
> We thank you for your insightful and valuable reviews and address your concerns as follows.
>
> **Q1**: What dataset is Vanilla RAG retrieving from? Is the info coming from the same KB but without MCTS reranking? And I think the setup for the Vanilla-RAG is not clear. It's later improved by Fig. 6 but this information comes too late.
> **A1**: Thank you for pointing this out. Yes, the retrieved examples come from the same knowledge base, excluding the reasoning context, as shown in Table 1. Besides, Vanilla-RAG uses the same hybrid retrieval module, which is the same as our RCTS. The difference is that it only relies on the feature similarity without MCTS reranking. And we will revise this explanation earlier about Vanilla-RAG in Section 3.1 and Section 4.2.
>
> **Q2**: L200-201: what is the score being used here? Why do you need SC if you have ground-truth answers?
> **A2**: As shown in Fig. 3(b), the score is defined as the ratio of correctly predicted answers to the total number of predicted answers, computed as:  $\text{Score} = \frac{N_{\text{correct}}}{N_{\text{total}}}$, where $N_{\text{correct}}$ = number of correct responses and $N_{\text{total}}$ = total number of responses.
> For subsequent question, SC serves to build a VQA knowledge base with reasoning contexts. While we possess both questiones and ground-truth answers from the knowledge base, the associated reasoning contexts are unavailable. Additionally, since the model-generated reasoning paths are not entirely reliable, we use the ground-truth to verify and score these reasoning contexts. Table 6 further demonstrates the reliability of our reasoning contexts.
>
> **Q3**: The purpose of rewards (especially the mutual consistency reward) should be made clear earlier on.
> **A3**: Thank you for pointing this out. We conduct more discussions about the purpose of rewards in Reviewer eQW2 **Q3**. And we will add this explantion earlier on Introduction (L88).
>
> **Q4**: How are you embedding images and text? Which model is used?
> **A4**: Following Lin et al. [1], we employ the Bert-base model (110M parameters) as our text encoder for generating text embeddings. For extracting image embeddings, we utilize a ViT-L coupled with a 2-layer MLP, totaling 312M parameters. We will add this in Section 4.2.
>
> **Q5**: Which model is used to construct the KB?
> **A5**: Thank you for pointing this out. We use Qwen2-VL (7B) to construct the KB. We will add this in L290.
>
> **Q6**: L262, does this mean that you are treating selecting a QA pair is an action? And it's not clear what the action space is or why this needs to be a sequential problem. In this case, the authors are selecting a set of ICL examples. It doesn't seem to me like order matters here, i.e. does selecting a different example first makes a difference?
> **A6**: As illustrated in Eq. 5, our action space comprises multiple retrieved QA pairs, where each QA pair represents a distinct action. Besides, this can be treated as a sequential problem because ICL performance depends critically not only on the quality of the retrieved samples but also on the their orders. Accordingly, our RCTS treats different example orders as sequential decision-making problems within a set of actions (L236). Furthermore, through comprehensive ablation studies, we systematically investigate and demonstrate the importance of example orders in Reviewer **x7Ua Q1**.
>
> **Q7**: Computational cost is mentioned in the limitations, it would be good to have a number to put to this.
> **A7**: We discuss the computational cost at Reviewer **yd6M Q2**.
>
> **Q8**: Essential references should be discussed.
> **A8**: Thank you for pointing these important references, we will cite them and add discussions in the revised version. We summarize the differences as follows:
> 1. LLM-R [2] introduces an iterative training framework to retrieve high-quality in-context examples for large language models. In contrast, our method employs a training-free MCTS-based reranking approach, offering greater generalization compared to trained methods.
> 2. RATP [3] leverages MCTS + RAG to enhance the self-reflection and self-critique capabilities across numerous private healthcare documents. While sharing a similar MCTS + RAG concept, there are differences in design details, such as the setup of proxy rewards and our heuristic rewards, as well as variations in the action space design. Additionally, RATP’s knowledge base consists of document-style data, whereas our method focuses on example pairs with reasoning contexts.
>
> **Q9**: A few small quibbles and typos about the language in the paper.
> **A9**: Thank you again for pointing this out, we will fix them in our revision.
>
> **Reference**
>
> [1] Lin et al., 2024, PreFLMR: Scaling up fine-grained late-interaction multi-modal retrievers
>
> [2] Wang et al., 2023, Learning to Retrieve In-Context Examples for Large Language Models
>
> [3] Pouplin et al., 2024, Retrieval Augmented Thought Process for Private Data Handling in Healthcare

---

### Official Review · Reviewer_eQW2 · 2025-03-14

**Overall Recommendation:** 3

**Summary:**

This paper presents a method to refine the selection of retrieved examples for multimodal language model in-context learning. It has two key components. The first component is to ask LLM to generate a set of rationale/reasoning contexts given a QA pair and select the context that has the highest probability of generating the answer given the question and context. The second component is to refine the selection of K candidates from an initial retrieved candidate set. It first select candidates based on the distribution from similarity scores, and then gradually update the distribution by checking 1) consistency between the answer generated from a selected candidate with the question and the actual answer from the selected candidate and 2) whether the answer generated from a selected candidate with the question can positively contribute to the prediction other questions. They conduct experiments on different benchmarks and find that the proposed method outperforms methods without retrieval, vanilla in-context learning with random examples, and vanilla retrieval.

**Update after Rebuttal.** The authors addressed most of my concerns, but my main concern about the knowledge base still partly remains because this work considers training data the knowledge base instead of traditional external knowledge. Although the authors use arguments such as dynamical change of the knowledge or generalization to different knowledge, these arguments are a bit weak considering that the knowledge candidates are large-scale training data. So I believe it is appropriate to keep my current score.

**Claims And Evidence:**

One major concern I have for this paper is the assumption of this paper. Unlike other KB-VQA work that mostly focuses on an external knowledge base (e.g., Wikipedia), the knowledge base in this paper is a large number of question-answer pairs for each question. Based on the large number of question-answer pairs, the authors propose a training-free algorithm to use those question-answer candidates as in-context examples to enhance open-source LLMs such as Qwen2-VL and InternVL-2. But it is not quite reasonable to me to not do some fine-tuning (either full parameter fine-tuning or PEFT), given the large number of examples for each task, while merely using them as candidates for in-context learning. I would expect the authors to provide more explanations on the motivation.

This paper can be much stronger if there is another setting to compare methods that are trained on this dataset. The authors can leverage this inference algorithm to select candidates on top of those fine-tuned models while showing it can still benefit from it.

**Essential References Not Discussed:**

N/A

**Experimental Designs Or Analyses:**

The experimental designs are legitimate. It clearly demonstrates the effectiveness of the proposed methods compared to methods without retrieval, vanilla in-context learning with random examples, and vanilla retrieval. And the ablation study also demonstrates the effectiveness of the use of MCTS and reasoning context.

**Methods And Evaluation Criteria:**

The methods of generating reasoning context make sense of this problem, and the motivation of using reranking methods to refine the candidates for answer generation is clear. The benchmarks are also legitimate.

But I would expect the authors to provide more explanations on the choices of the two heuristics rewards. For self-consistency rewards, it promotes the selection of QA pairs that the answer matches the generated answer. But what is the rationale behind this heuristics?

**Other Comments Or Suggestions:**

No

**Other Strengths And Weaknesses:**

This paper is well-structured and easy to follow.

**Questions For Authors:**

No

**Relation To Broader Scientific Literature:**

This paper can provide a paradigm for improving the quality of multimodal examples when conducting multimodal in-context learning.

**Theoretical Claims:**

N/A

---

> ### Author Rebuttal · Authors · 2025-04-01
>
> We thank you for your constructive reviews and address your concerns as follows.
>
> **Q1**: One major concern is the assumption of this paper. Unlike other KB-VQA that mostly focuses on an external knowledge base, the knowledge base in this paper is a large number of QA pairs for each question.
> **A1**: In RCTS, we propose a novel paradigm with reasoning VQA samples serves as the knowledge base, different from traditional RAG that relies on external sources (e.g., Wikipedia or Online Search), as detailed in L34-40.Specifically, our approach leverages reasoning QA pairs to explicitly model ICL, enhancing the model's ability to solve complex problems.
> To address potential concerns about the source of KB, we leverage multiple open-source VQA datasets (e.g., ScienceQA, OK-VQA, VizWiz) and further employ an automated reasoning construction framework to build a high-quality reasoning KB.
> In addition, regarding concerns about the assumptions in this scenario, RCTS aims to transition VLMs’ responses from merely known to better understood (know-how reasoning) by prepending it with retrieved reasoning examples, extensive experiments (Section 4.3) demonstrate the effectiveness of our approach, achieving a significant improvement on complex reasoning benchmarks compared to Vanilla-RAG baselines.
>
> **Q2**: It is not quite reasonable to me to not do some fine-tuning, given the large number of examples for each task, while merely using them as candidates for in-context learning. I would expect the authors to provide more explanations on the motivation.
> **A2**: Thank you for raising this important concern. Actually, the core innovation of our paper lies in exploring a method that leverages massive reasoning QA pairs as a knowledge base for performance improvement without requiring training. We will provide more explanations motivated by three key considerations:
>   1. Generalization Capability
> Although fine-tuning methods (SFT/PEFT) are effective, the debate between SFT and ICL hinges on a trade-off between specialization and generalization. While SFT offers more tailored and often higher performing models for specific tasks, it can lead to loss of the model’s generalization abilities, as discussed by Chen et al.[1]
>   2. Flexible Knowledge Base Construction
> Compared to fine-tuning, our framework is training-free and can be adaptively extended to multiple domains by simply expanding the knowledge base, as mentioned in L161-164, offering greater universality. Additionally, we have added this explanation in our Introduction to further emphasize our motivation.
>   3. Experiment Validation
> We follow the suggestions and explicitly compared RCTS vs. fine-tuning variants. Specifically, we take VQA pairs from the knowledge base corresponding to MathV as training samples and use Llama-Factory[2] for SFT on Qwen2-VL-2B. As shown below, the results showed that while fine-tuning improved accuracy on our test set, it caused a significant drop on out-of-domain benchmarks, validating our design choice. We believe this trade-off favors applications requiring broad adaptability.
>
> |Method|MathV|ScienceQA|
> |-|-|-|
> |Zero-Shot|18.75|67.18|
> |Fine-tuning-on-MathV-1epoch|22.69|44.56 (OOD)|
> |Fine-tuning-on-MathV-3epoch|23.03|43.67 (OOD)|
> |RCTS (ours)|22.04|78.99|
>
> **Q3**: Expect the authors to provide more explanations on the choices of the two heuristics rewards. For self-consistency rewards, it promotes the selection of QA pairs that the answer matches the generated answer. But what is the rationale behind this heuristics?
> **A3**: Regarding the self-consistency reward, we primarily leverage the self-consistency property of VLM models (Wang et al.[3]), i.e., 'Self-consistency leverages the intuition that a complex problem typically admits multiple different ways of reasoning path leading to its unique answer.' Thus, the essence of selecting generated answers $\{A_i^{(n)}\}$ (L265) that match the predicted answers $\tilde{A}_i$ (L266) lies in choosing responses where the predicted answers and the predicted reasoning paths remain consistent.
> For the mutual heuristic reward, we posit that if the answer to one question is correct, it will positively contribute to reasoning for other related questions, and vice versa (L277-280). Specifically, the reward is based on whether the reasoning context generated by MCTS-HR can generalize effectively to other questions (from KB), thereby selecting robust and transferable response.
> Besides, we comprehensively take into account the two aforementioned rewards and conducted thorough ablation experiments to verify their effectiveness as shown in table 5.
>
> **Reference**
>
> [1] Chen et al., 2020, ACL, Recall and Learn: Fine-tuning Deep Pretrained Language Models with Less Forgetting
>
> [2] Zheng et al., 2024, ACL, LlamaFactory: Unified Efficient Fine-Tuning of 100+ Language Models
>
> [3] Wang et al., 2023, ICLR, Self-Consistency Improves Chain of Thought Reasoning in Language Models

---

> > ### Comment · Reviewer_eQW2 · 2025-04-03
> >
> > Thanks for the authors' comments. But I still have concerns about the fine-tuning and knowledge utilization parts.
> >
> > The authors explained that their method demonstrates better generalization across different domains. Although I agree with the authors that this training-free methods show good performance on both MathV and ScienceQA in the above table, I cannot ignore that the cost here is that we have a large "training set" merely for retrieval purposes. When we have a large training set, it kind of weakens the argument regarding the out-of-distribution. Because we just have a lot of in-domain data that could have been used to fine-tune the model rather than just serving as the candidates.

---

> > > ### Author Response · Authors · 2025-04-09
> > >
> > > We appreciate the reviewers' thoughtful feedback and acknowledging our perspective about generalization capabilities. We will reply to your outstanding concern in the order as follows:
> > >
> > > 1. Regarding the concern of the fine-tuning parts, actually, the decision to fine-tune a model depends significantly on the suitability of the approach for specific application scenarios. In cases where the knowledge base is static and abundant Visual Question Answering (VQA) samples are available, fine-tuning the model with the knowledge base proves to be a better strategy. This is supported by the empirical results presented in our earlier response. Nevertheless, our approach, which takes advantage of retrieved reasoning contexts, can be applied to solve more flexible and open-ended scenarios, i.e., those characterized by limited training resources and the need for frequent, dynamic updates to the knowledge base. These scenarios demand a more adaptable solution that can meet evolving requirements without relying on extensive retraining.
> > >
> > > 2. Regarding the unknown distribution of the test data, where it is impossible to predict the specific problems a model may encounter, our approach provides a more practical solution for domain-specific needs. Specifically, one can construct a personalized knowledge base within their target domain and then leverage frozen Vision-Language Models (VLMs) to improve response reasoning and overall performance. Moreover, as our RCTS operates without the need for additional fine-tuning, it is particularly suited for customized RAG applications. This leads to a key practical advantage our approach: it enables multiple users to share a single deployed vision-language model. By maintaining a personalized knowledge base for each user, our method achieves rapid customization without incurring additional training and deploying costs. This not only reduces computational overhead but also significantly cuts down on deployment costs.
> > >
> > > 3. Regarding the reviewers' concerns about knowledge utilization and the associated cost, actually, our method emphasizes reasoning-context knowledge rather than an reliance on excessively large knowledge base. By employing our MCTS-HR re-ranking strategy, we could achieve stronger performance by focusing on the patterns of "relevant" examples. Specifically, "relevant" here is distinct from mere "similar". The key to our RCTS lies in identifying contextually appropriate knowledge, ensuring that the focus remains on quality and relevance rather than relying on the scale of the knowledge base.
> > >
> > > We hope this clarification helps to better align our method focus with the reviewers’ expectations, and we will provide a more detailed explanation of the motivations behind our proposed approach in the revision.

---

### Official Review · Reviewer_x7Ua · 2025-03-21

**Overall Recommendation:** 3

**Summary:**

This work proposed a multi-modal RAG method to retrieve reasoning context examples from the knowledge base. The main components of this work are (1) a CoT knowledge base, (2) knowledge retrieval metrics with hybrid vision-language embeddings, and (3) Monte Carlo Tree Search (MCTS) to retrieve the most related examples. Experiments on several benchmarks and ablation studies demonstrate the effectiveness of the components and the pipeline.

**Claims And Evidence:**

The claims are supported by clear and convincing evidence.

**Essential References Not Discussed:**

Currently no.

**Experimental Designs Or Analyses:**

The overall experimental designs look good to me.

**Methods And Evaluation Criteria:**

The evaluation makes sense.

**Other Comments Or Suggestions:**

Please see Weaknesses.

**Other Strengths And Weaknesses:**

Strengths:

* The performance of the proposed framework looks strong. It outperforms baseline methods by large margins on several benchmarks. The ablations on the components are relatively comprehensive.

* The overall writing of this paper is clear. The figures and tables are well organized and designed.

* The ideas of reasoning knowledge base and hybrid embeddings for retrieval make sense.

Weaknesses:

* My biggest concern is on the complexity of the framework, especially the application of MCTS. In this work, MCTS is used to select most related examples. However, it is not clear to me why in-context example retrieval needs such a complex tree-based search strategy, especially for an ordered chain of examples. Does the order of retrieved examples matter? Are there any observations on the chain structure, like how the model selects the examples in such orders? The application of MCTS looks effective, but the inspiration behind the application is not strong to me. Also, the time computation cost is not provided, and it is not clear whether the overall framework has a heavy time cost.

* Although the other two components, reasoning knowledge base and hybrid embeddings, make sense and work positively to the final performance, the novelty and differences of these two modules compared to the references are not clear.


--------------


The rebuttal addressed my major concern on the impact of example order and the necessity of MCTS. Therefore, I am happy to increase my score.

**Questions For Authors:**

Please see Weaknesses.

**Relation To Broader Scientific Literature:**

I think no.

**Theoretical Claims:**

This work does not include theoretical proofs.

---

> ### Author Rebuttal · Authors · 2025-04-01
>
> We thank you for your constructive reviews and address your concerns as follows.
>
> **Q1**: Why in-context example retrieval needs such a complex tree-based search strategy? Does the order of retrieved examples matter?
> **A1**: As demonstrated by Tan et al.[1] and Liu et al.[2], the order of retrieved in-context examples has an important effect on the generative model's response due to path dependency properties, i.e. different example orders implicitly guide the model's attention toward distinct reasoning patterns. Recognizing both the importance of example order and the combinatorial complexity in identifying optimal order, we employ a tree-based search strategy for re-ranking. This approach effectively balances exploration of potential orders with exploitation of high-performing orders through its hierarchical search structure. Besides, we further conduct experiments with shuffle/ordered examples on ScienceQA dataset to empirically validate the impact of order, with the results shown below:
>
> |Model|Retrieval-Order|Retrieval-Shuffle|RCTS-Shuffle|RCTS (ours)|
> |-|-|-|-|-|
> |Qwen2-VL-2B|71.94|71.26|73.90|78.99|
> |Qwen2-VL-7B|86.68|87.95|88.23|91.44|
> |InternVL2-8B|93.00|92.57|93.16|94.20|
>
> Where, 'Retrieval-Order' refers to the top-3 samples sorted by similarity (i.e., Vanilla-RAG), 'Retrieval-Shuffle' denotes randomly reordered top-3 retrieval samples. 'RCTS-Shuffle' represents the shuffled orders of our RCTS re-ranking. The results demonstrate that different orders exert influence on model's performance.
>
> **Q2**: The complexity of the framework, especially the application of MCTS. And the inspiration behind the application of MCTS.
> **A2**: We justify our adoption of MCTS from the following aspects:
> 1. As mentioned in Q1, the order of the examples is important, which motivates us to optimize the example order. Similar to path planning (Eiffert et al.[3]), this problem can be modeled as a sequential searching problem, which is very suitable to be solved using MCTS.
> 2. MCTS effectively balances exploration of potential orders with exploitation of high-performing orders through its hierarchical search structure, making it more efficient than brute-force-search‌‌. Besides, our approach produce substantial performance gains, clearly justifying the computational investment in MCTS.
>
> Therefore, we adopt MCTS over conventional retrieval methods to transition from exploiting "similar examples" to "relevant examples" (L28).
>
> **Q3**: Are there any observations on the chain structure, like how the model selects the examples in such orders?
> **A3**: Through observation of some examples of our RCTS in supplementary materials , we summarize some patterns below:
> 1. In the initial expansion of RCTS, RCTS prioritizes expanding from the most similar examples. When the reward value Q is high, RCTS assigns higher priority (Fig. 15). However, if the similar example face significant degradation (i.e., have very low reward values), RCTS will exclude it (Fig. 14).
> 2. During the expansion process of RCTS, as shown in Fig. 11 and Fig. 12, valuable examples (i.e., those with higher reward values) are repeatedly explored and utilized, even if they appear in different orders. In contrast, examples with low reward values are only used once and are not revisited afterward.
> 3. After RCTS reaches the maximum number of simulation rounds, it selects the branch with the highest cumulative reward value Q as the final result, even though other branches may also contain correct answers.
> Note: For all examples in the supplementary materials, the branch expansion follows a left-to-right sequence.
>
> **Q4**: The time computation cost.
> **A4**: We discuss this concern at Reviewer **yd6M Q2**.
>
> **Q5**: The novelty and differences of reasoning knowledge base and hybrid embeddings compared to the references.
> **A5**: 1. For reasoning knowledge base, while both our approach and AutoCoT[4] automate the generation of reasoning chains, our key novelty lies in proposing self-consistency strategy to verify the correctness of the reasoning context, making it suitable for RAG applications.  We elaborate on this motivation in L185–190. 2. For hybrid embeddings, we directly adopt the retrieval module (frozen) from PreFLMR[5] in L211. The core contributions of this work lie in making them suitable with reasoning knowledge base and an innovative tree-based search strategy for re-ranking examples.
>
> **Reference**
>
> [1] Tan et al., 2023, TACL, Lost in the Middle: How Language Models Use Long Contexts
>
> [2] Liu et al., 2024, Order Matters: Exploring Order Sensitivity in Multimodal Large Language Models
>
> [3] Eiffert et al., 2020, ICRA, Path Planning in Dynamic Environments using Generative RNNs and Monte Carlo Tree Search
>
> [4] Zhang et al., 2023, ICLR, Automatic chain of thought prompting in large language models
>
> [5] Lin et al., 2024, ACL, PreFLMR: Scaling up fine-grained late-interaction multi-modal retrievers

---

### Decision · Program_Chairs · 2025-05-01

**Decision:**

Accept (spotlight poster)

**Comment:**

**Summary**
This paper proposes a multimodal RAG framework to solve the limitation of instruction misalignments. The authors present empirical study and validate their approach on several dataset such as ScienceQA and MMMU-Dev.

**Reviewer Consensus and Review Quality**
The submission received four reviews. All reviewers provided positive feedback, highlighting the novelty and good empirical results.

**Strengths**
- The novelty is good.
- The results of the paper are strong, with consistent gains across domains
- The ablations on the components are relatively comprehensive.
- The overall writing of this paper is clear.
- The figures and tables are well organized and designed.

**Weaknesses**
- weak experiments to demonstrate the capability of this framework to generalize the external knowledge

**Justification**
Based on the reviewers comments and the author’s response, I believe this paper clearly contributes to the community of multimodality learning. While there exist a few minor issues, such as the complexity of the framework, the strengths of the paper outweigh its weaknesses significantly. In addition, it ranks 1 among 12 papers. Therefore, I recommend **strong accept**.